

# Four dimensional variational inversion of black carbon emissions during ARACTAS-CARB with WRFDA-Chem

Jonathan J. Guerrette[1] and Daven K. Henze[1]

[1]Department of Mechanical Engineering, University of Colorado, Boulder, CO, 80309, U.S.A.

*Correspondence to:* Jonathan Guerrette (jonathan.guerrette@colorado.edu)

**Abstract.** Biomass burning emissions of atmospheric aerosols, including black carbon, are growing due to increased global drought, and comprise a large source of uncertainty in regional climate and air quality studies. We develop and apply new incremental 4D-Var capabilities in WRFDA-Chem to find optimal spatially and temporally distributed biomass burning (BB) and anthropogenic black carbon (BC) aerosol emissions. The constraints are provided by aircraft BC concentrations from the Arctic Research of the Composition of the Troposphere from Aircraft and Satellites in collaboration with the California Air Resources Board (ARCTAS-CARB) field campaign and surface BC concentrations from the Interagency Monitoring of PROtected Visual Environment (IMPROVE) network on 22, 23, and 24 June, 2008. We consider multiple BB inventories, including Fire INventory from NCAR (FINN) v1.0 and v1.5 and Quick Fire Emissions Database (QFED) v2.4r8. On 22 June, aircraft observations are able to reduce the spread between QFED$\times\frac{1}{3}$ and FINNv1.0 from $\times 3.5$ to $\times 2.1$. On 23 and 24 June, the spread is reduced from $\times 3.4$ to $\times 1.4$. The posterior corrections to emissions are heterogenous in time and space, and exhibit similar spatial patterns of sign for both inventories. The posterior diurnal BB patterns indicate that multiple daily emission peaks might be warranted in specific regions of California. The U.S. EPA's 2005 National Emissions Inventory (NEI05) is used as the anthropogenic prior. On 23 and 24 June, the coastal California posterior is scaled by $\times\frac{1}{2}$, where highway sources dominate, while inland sources are increased near Barstow by $\times 5$. Relative BB emission variances are reduced from the prior by up to 35% in grid cells close to aircraft flight paths and up to 60% for fires near surface measurements. Anthropogenic variance reduction is as high as 40% and is similarly limited to sources close to observations. We find that the 22 June aircraft observations are able to constrain approximately 14 degrees of freedom of signal (DOF), while surface and aircraft observations together on 23/24 June constrain 23 DOF. Improving hourly to daily scale concentration predictions of BC and other aerosols during BB events will require more comprehensive and/or targeted measurements and a more complete accounting of sources of error besides the emissions.

## 1 Introduction

Black carbon (BC) makes significant contributions to short term climate (Bond et al., 2013) and human health (Janssen et al., 2012) as a component of aerosolized fine particulate matter ($PM_{2.5}$) in the atmosphere. BC is emitted through incomplete combustion from natural and anthropogenic burning of biomass and fossil fuels. Open biomass burning (BB), which includes natural wild fires, deforestation, and agricultural waste and prescribed burning, accounts for 40% of total global BC emissions,



while anthropogenic energy related sources (e.g., on- and off-road diesel and gasoline engines, industrial coal, residential cooking and heating) make up the remaining 60% (Bond et al., 2013). Future climate conditions that increase drought and fire prevalence (e.g., Spracklen et al., 2009) and increasingly regulated anthropogenic sources might lead to a reversal of these ratios in California (Mao et al., 2011) and globally (Jolly et al., 2015). In California, BB events have been shown to increase

surface $PM_{2.5}$ concentrations by $\times 3$ to $\times 5$, compared to non-fire periods (Wu et al., 2006). The heterogeneity in BC emission and loss patterns and difficulty in replicating transport contribute to prediction uncertainty.

Despite the recognized importance of biomass emissions, large discrepancies remain in inventories in terms of biomass consumed and emitted chemical species. Zhang et al. (2014a) considered seven different inventories during February 2010 over Africa, which gave a range of $\times 12$ in total emitted OC and BC throughout the month. Fu et al. (2012) found similar

variability between only two inventories in Southeast and East Asia during January and April 2006. Zhang et al. (2014a) concluded that diffusion and loss mechanisms limit the effect of the monthly emission variability to only a $\times 2$-3 range of monthly average domain-wide burden, AOD, and 2 m temperature. However, the inventory spread of emission magnitudes from larger sources led to column burden ranges of $\times 16$-30 at the hourly to daily grid scales.

The large range in inventories at small scales results from the differing ways in which they are built. In order to be glob-

ally applicable, fire locating algorithms use remotely sensed hot spots from polar-orbiting satellites. Some provide additional regional locational and diurnal information with geostationary instruments. In all cases, daily emissions in a grid cell are calculated as the product of activity (kg burned) and emission factors for each species and vegetation class combination (kg emitted (kg burned)$^{-1}$). Bottom-up inventories combine rough estimates of burned area with vegetation densities and percent biomass burned associated with different Land Cover Types (LCT) to determine fire activity (e.g., Wiedinmyer et al.,

2011; Reid et al., 2009; van der Werf et al., 2010). Top-down approaches use fire radiative power (FRP) measured by polar-orbiting or geostationary satellites and the LCT-specific energy content (e.g., Kaiser et al., 2012; Zhang et al., 2012), which circumvents using uncertain estimates of burned areas (Boschetti et al., 2004). A third approach combines the FRP with top-down constraints of aerosol optical depth (AOD) (e.g., Ichoku et al., 2012; Darmenov and da Silva, 2013). All three of these approaches cross reference fire locations with biome lookup tables to obtain the species-specific emission factors for each fire.

Improving short-term, local BC concentration predictions requires characterizing fine-scale spatial and diurnal patterns of BB emissions. The weakness of using only polar-orbiting data (e.g., Moderate Resolution Imaging Spectroradiometer (MODIS) instruments aboard Terra and Aqua) in bottom-up fire inventories is that there are nominally four overpasses per day, often with missed detections due to cloud- and smoke-cover or fire sizes beyond the instrument detection limits. Thus, these observations provide little information about the diurnal pattern of fire counts and FRP. Zhang et al. (2012) and Andela et al. (2015) devise

methods for deriving climatological diurnal FRP patterns using geostationary observations. Both provide new information to modelers, but the former is not generalizable to grid-scale diurnal variability and the latter precludes the possibility that diurnal FRP (Zhang et al., 2012) and emissions (Saide et al., 2015) patterns may be bimodal for specific LCT's and fire regimes, or due to local meteorology.

In contrast to their BB counterparts, anthropogenic emissions of BC are periodic across weekly and annual time scales. Their

spatial distributions are relatively well-known in developed countries, and less so in developing countries (Bond et al., 2013).



Global estimates of annual anthropogenic BC emissions vary by $\times 2$ (Bond et al., 2013), national annual BC emissions in Asian countries and regions have uncertainties from $\times 2$ to $\times 5$ (Streets et al., 2003). In North America, including in California, uncertainties still persist in terms of characterizing the magnitude of emissions in a particular year, seasonal variability, and long term trends in activity and control strategies (Grieshop et al., 2006; McDonald et al., 2015). Bond et al. (2013) cite several

inventories that give a range of $\times 1.7$ for annual U.S. anthropogenic BC emissions. However, like many other inventories, the U.S. EPA National Emission Inventory (Reff et al., 2009) does not specify uncertainty bounds either for the whole country or at state and county levels.

These challenges in characterization of both BB and anthropogenic emissions of BC and co-emitted species have led to the proliferation of top-down constraint methods of varying complexity and utility. Several studies have used adjoint-free methods

for anthropogenic emissions in Los Angeles, California using aircraft measurements during the 2010 California Research at the Nexus of Air Quality and Climate Change (CalNex) campaign. Brioude et al. (2012) constrained CO, $NO_x$, and $CO_2$, and Cui et al. (2015) constrained $CH_4$; both applied a Lagrangian Particle Dispersion Model (LPDM). Peischl et al. (2013) constrained $CH_4$ using a mass balance approach and light alkane signatures from multiple sectors. LPDM benefits from being able to resolve sources on as fine of a grid resolution as is used in the underlying model. Both LPDM and mass balance are limited

to linear tracer problems where observations are recorded under specific meteorological conditions. Wecht et al. (2014) used GEOS-Chem in an analytical inversion to compare constraints from the CalNex aircraft measurements with those from present and future satellite observations of $CH_4$ throughout California. Although an analytical inversion does not require an adjoint, the approach is limited, computationally, to constraining only a few sources, which imposes aggregation error (Mao et al., 2015). Adjoint-based four-dimensional variational data assimilation (4D-Var) is able to account for nonlinear behavior between the

emission sources and observation receptors by calculating exact gradients across physical processes. Such an approach does not have the limitations imposed by mass balance, LPDM, or analytical inversions, but does require development of an adjoint. The gradients are usually calculated through an adjoint model, although recent work (Saide et al., 2015) performs 4D-Var on a limited area fire without an adjoint. That new approach, while easier to implement, is limited to solving for only a few spatially-distributed sources due to computational limitations.

In this study, we adapt the adjoint-based incremental four dimensional variational data assimilation (incremental 4D-Var) used in the WRFDA weather forecasting system (Barker et al., 2005; Huang et al., 2009) to solution of tracer surface flux estimation problems. We apply the resulting tool, WRFDA-Chem, to constrain anthropogenic and BB sources of BC throughout California during the Arctic Research of the Composition of the Troposphere from Aircraft and Satellites in collaboration with the California Air Resources Board (ARCTAS-CARB) field campaign. In June 2008, ARCTAS-CARB characterized aerosols

and trace gases throughout California with DC-8 aircraft flights on 20 (Friday), 22 (Sunday), 24 (Tuesday), and 26 (Wednesday) June (Jacob et al., 2010). Sahu et al. (2012) used BC total mass measurements from a single-particle soot photometer (SP2) and other simultaneous gas-phase measurements to identify and characterize anthropogenic and BB plumes in California. We assess the capability of these same observations and every-3rd-day surface measurements from the Interagency Monitoring of PROtected Visual Environment (IMPROVE) network to constrain errors in BC surface fluxes when used in 4D-Var. As

described in (Guerrette and Henze, 2015), this approach of assimilating chemical tracer observations in a regional numerical




weather prediction and chemistry model is unique in the context of previous 4D-Var flux constraints. We also estimate emissions, their associated uncertainties, and provide diagnostics for observing system evaluation at high spatio-temporal resolution (hourly, $18\,\mathrm{km} \times 18\,\mathrm{km}$). The approach taken in this work is described in Sec. 2, including the forward, adjoint, and tangent linear models, the prior inventories and domain, and the incremental 4D-Var method implemented in WRFDA-Chem. Section 3

describes the application of WRFDA-Chem to the BB and anthropogenic emission inversion problem during ARCTAS-CARB. We conclude with a summary and recommendations for future measurement campaigns and emission inversion research.

## 2 Method

### 2.1 Nonlinear, adjoint, and tangent linear models

Incremental 4D-Var requires forward nonlinear (NLM), adjoint (ADM), and tangent linear (TLM) models. The NLM is nearly

identical to WRF-Chem (Grell et al., 2005) with the addition of emission scaling factors. The GOCART option facilitates 19 species, including 4 gas and aerosol species for sulfate chemistry, hydrophobic and hydrophilic BC and organic carbon, 5 size bins for dust, 4 bins for sea salt, and 2 diagnostic species for $PM_{2.5}$ and $PM_{10}$. While we use GOCART, the results presented are limited to BC. The model configuration is the same as was used in Guerrette and Henze (2015), and is summarized as follows: ACM2 PBL mixing (Pleim, 2007a, b), Pleim-Xiu land surface model (Xiu and Pleim, 2001; Pleim and Xiu, 2003;

Pleim and Gilliam, 2009) and surface layer (Pleim, 2006) mechanisms without soil moisture and temperature nudging, Wesely dry deposition velocities (Wesely, 1989), GSFC shortwave and Goddard long wave radiation, and microphysics turned off. Microphysical and radiative responses to online aerosols are also turned off, because they are not included in WRF-Chem for GOCART.

We utilize the recently developed WRFPLUS-Chem (Guerrette and Henze, 2015), which contains ADM and TLM code

extending the original WRFPLUS software (Zhang et al., 2013). WRFPLUS-Chem describes chemical tracers in the context of planetary boundary layer (PBL) mixing, emissions, dry deposition, and GOCART aerosols. ADM and TLM gradients have been verified against finite difference approximations. Second-order checkpointing reduces the memory footprint to a feasible level for ADM and TLM simulations over longer durations ($>\sim 6$ hr) and/or that use many chemical tracers ($>\sim 10$). Guerrette and Henze (2015) applied the ADM in calculating sensitivities relevant to the emission inversion carried out here.

Sec. 3.5 includes a comparison of the results of that study with the posterior emissions here.

The model domain is similar to that used by Guerrette and Henze (2015). The spatial extent encompasses California and other southwest U.S. states. We conduct two emission inversions, the first on 22 June with a focus on biomass burning sources, and the second on 23-24 June with a focus on anthropogenic sources. We generated chemical initial conditions by running WRF-Chem from 15 June 2008, 00:00:00 up until the beginning of each inversion period. We used the default WRF-Chem

boundary condition for BC concentration of $0.02\,\mu\mathrm{g\,kg^{-1}}$, which was found to be consistent with observations with an upwind flight on 22 June. Meteorological initial and boundary conditions are interpolated from $3\,\mathrm{h}$, $32\,\mathrm{km}$ North American Regional Reanalysis (NARR) fields. The horizontal resolution is $18\,\mathrm{km}$ throughout $80{\times}80$ columns, and there are 42 vertical levels between the surface and model top at $100\,\mathrm{hPa}$.





## 2.2 Prior emission inventories

The prior includes sources of BC from anthropogenic activity and natural wild fires. Anthropogenic emissions are taken from the U.S. EPA's 2005 National Emissions Inventory (NEI05) for mobile and point sources, including for example diesel on-road and power production from coal. The individual sectors are lumped together for each grid cell. We represent BB emissions using three different wild fire inventories, FINNv1.0 and v1.5 both at 1 km × 1 km resolution (Wiedinmyer et al., 2011, 2006), and QFEDv2.4r8 at $0.1° × 0.1°$ resolution (Darmenov and da Silva, 2013). FINNv1.5 is readily available through NCAR (http://bai.acom.ucar.edu/Data/fire/) to WRF-Chem users, while FINNv1.0 is no longer supported. However, we include FINNv1.0 in this study, because it shows equivalent value as a prior. FINN and QFED fall into the first (bottom-up) and third (top-down constraint with AOD) category of BB inventories described in Sec. 1, respectively. QFED scales global aerosol emissions from four biome types through multiple linear regression between observed MODIS aerosol optical depth (AOD) and modeled GEOS-5 AOD during the years 2004-2009. For temperate forests, which produce 80% of the wild-fire BC in California during this modeling study (June 2008), QFED scales aerosols by ×4.5 throughout the world. This global scaling is problematic for the California fires, because the GEOS-5 AOD is biased high in the Western U.S. during the summer fire seasons of 2006-2008 (Fig. C14 of Darmenov and da Silva, 2013). In order to match the regional climatological AOD scaling factors for the Western U.S., we scale all QFED BC sources by $×\frac{1}{3}$. This scaling is already taken into account in the prior emissions shown in Sec. 3.3.

The WRF preprocessor distributed with the FINN inventory is used to distribute ASCII formatted lists of both FINN and QFED daily speciated fire emissions to hourly netcdf files readable by WRF. The diurnal profile follows the Western Regional Air Partnership profile WRAP (2005), and is defined by a flux peak from 13:00 to 14:00 Local Time (LT), and flat fluxes equal to 2.5% of the peak value between 19:00 LT and 09:00 LT. Through modeling experience, we found two bugs with how the FINN preprocessor interprets the WRAP profile and have fixed them for this case study. The total FINNv1.0 emissions across the model domain before and after fixing these bugs are plotted in Fig. 1 along with MODIS active fire counts (NASA). The first bug relates to how the timezone of a particular fire is calculated from longitude. The preprocessor converts a decimal longitude to integer time zone bins; this allows a fire at 120.1°W to be an hour earlier in the diurnal profile than a fire at 119.9°W, even though they should be at nearly identical positions in the WRAP profile. Such behavior might apply to anthropogenic emissions, where cities near time zone borders follow different daily cycles of activity, but not to natural activity related to the 15° per hour cycle of the sun.

The second bug, and the one most visible in Fig. 1, is in the redistribution of UTC fire detections in to LT emissions. MODIS Terra and Aqua overpass times are distributed around noon and midnight LT globally, with some adjustment as the image capture location moves farther from the equator. The fire hot spots are detected in UTC days, and their emissions are profiled according to LT periods corresponding to the same UTC day as the detection. In California, where the LT is UTC minus 8 hours, the noon overpass corresponds to 20:00 UTC, and 00:00 UTC corresponds to 16:00 LT on the previous day (sun cycle). Therefore, when a fire is detected during nearly peak heat and emission fluxes at noon, a large fraction of the flux is apportioned to the previous afternoon. For locations east of the International Date Line, the LT reallocation is in the opposite direction. In





either case, some portion of the profile is shifted by 24 hours. This error is apparent as a temporal discontinuity in the case of transient fires that vary significantly in magnitude from one day to the next, especially after a recent ignition. Since the domain used here is nearly confined to a single time zone, we simply move the emissions forward one day for times between 16:00-23:00 LT (00:00-07:00 UTC). A more robust fix will need to be implemented in a future preprocessor.

Another error in the prior BB emissions is less easily resolved. Figure 2 shows where the MODIS active fires are located relative to the inventory fire locations. Since QFED fires are provided on a LAT-LON grid, the fire centers do not coincide with its grid centers. When the inventory is distributed to the 18 km model grid, some emissions are shifted over by one column relative to the FINN locations. There are several additional spurious emission locations in QFED, where no active fires were detected on either 21 or 22 June. In a month-long simulation, differences in fire gridding between several inventories can be

averaged out. In the shorter term inversions over California presented in Sec. 3, the locational differences do affect the results.

## 2.3  WRFDA-Chem inversion system

The aim of data assimilation (DA) is to optimally combine uncertain observations with uncertain model predictions to provide an improved estimate of the state of a system than either gives alone. In Bayesian statistics, the probability distribution of a set of control variables (CV), $\boldsymbol{x} \in \mathcal{R}^n$, conditional on available observations, $\boldsymbol{y}^o$, is proportional to the product of two known

distributions,

$$P\left(\boldsymbol{x}|\boldsymbol{y}^o\right) \propto P\left(\boldsymbol{x}\right) P\left(\boldsymbol{y}^o|\boldsymbol{x}\right). \tag{1}$$

The first distribution on the right hand side is called the prior, background, or first guess; the second is the likelihood of model-observation mismatch, where here both are assumed to be Gaussian. They are found through the solution of the minimization problem

$$\min_{\boldsymbol{x}} \quad J\left(\boldsymbol{x}\right) = \frac{1}{2}\left(\boldsymbol{x} - \boldsymbol{x}_b\right)^\top \mathbf{B}^{-1}\left(\boldsymbol{x} - \boldsymbol{x}_b\right)$$

$$+ \frac{1}{2}\left(G\left(\boldsymbol{x}\right) - \boldsymbol{y}^o\right)^\top \mathbf{R}^{-1}\left(G\left(\boldsymbol{x}\right) - \boldsymbol{y}^o\right), \tag{2}$$

where $\boldsymbol{x}_b$ is the vector of prior CVs, $\mathbf{B}$ is the background covariance matrix, and $\mathbf{R}$ is the model-observation error covariance matrix. The nonlinear operator,

$$G\left(\boldsymbol{x}\right) = \begin{pmatrix} H_1\left(\boldsymbol{x}\right) \\ \vdots \\ H_i\left(\boldsymbol{x}\right) \\ \vdots \\ H_N\left(\boldsymbol{x}\right) \end{pmatrix}, \tag{3}$$

is similar to that applied by Weaver et al. (2005) and Tshimanga et al. (2008), and is composed of the model-observation

operators, with each $H_i$ mapping $\boldsymbol{x}$ to observation time $i$. The measurements at each acquisition time, $y_i^o \in \mathcal{R}^{m_i}$, are expressed





independently for $N$ acquisition times by

$$\boldsymbol{y}^o = \left[\boldsymbol{y}_1^{o\top}, \ldots, \boldsymbol{y}_N^{o\top}\right]^\top \in \mathcal{R}^m, \tag{4}$$

where $\sum_{i=1}^N m_i = m$. The $o$ superscript denotes that $\boldsymbol{y}^o$ are observations.

The cost function in Eq. 2 is derived for unbiased Gaussian statistics in both the background errors and model-observation errors. When grid-scale CV uncertainties are greater than 100%, as is often the case for chemical emissions, that assumption allows the posterior to be either positive or negative. While net surface flux rates can be negative when accounting for upward and downward rates together, emission rates are themselves positive. To ensure this, the ratio of modeled (posterior, $E_a$) to tabulated inventory (prior, $E_b$) emissions in all grid cells are gathered into a vector, $\boldsymbol{\beta} = e^{\boldsymbol{x}_a}$, such that

$$E_{a,j} = E_{b,j}\beta_j, \tag{5}$$

for CV member $j$. Each $\beta_j$ is a linear scaling factor, while exponential scaling factors comprise the posterior CV vector, $\boldsymbol{x}_a$. Fletcher and Zupanski (2007) showed that this approach – which was previously utilized in emission inversions by, e.g., Müller and Stavrakou (2005), Elbern et al. (2007), and Henze et al. (2009) – converges toward the median of a multivariate log-normal distribution for $\boldsymbol{\beta}$. Although other emission scaling forms have proven effective (Bergamaschi et al., 2009; Jiang et al., 2015), we stick with exponential scaling factors here both as a first demonstration, and to be consistent with log-normal statistics for emission rates. $\boldsymbol{x}$ is resolved on the grid scale and across hourly discretized emission rates; the temporal resolution is customizable.

### 2.3.1 Incremental 4D-Var

Here we apply incremental 4D-Var as first introduced by Courtier et al. (1994), utilizing the existing software architecture in WRFDA, and extended to accommodate exponential emission scaling factor CVs. Incremental 4D-Var starts from the assumption that model evaluations of perturbed CVs at observation time $k$ can be expressed by

$$G\left(\boldsymbol{x} + \delta\boldsymbol{x}\right) \approx G\left(\boldsymbol{x}\right) + \mathbf{G}\delta\boldsymbol{x}, \tag{6}$$

where $\mathbf{G}$ is the Jacobian. The full matrix is too large to store in memory, but the product $\mathbf{G}\delta\boldsymbol{x}$ is found through the TLM, transforming increments in CV space to perturbations in observation space. With the assumption, Eq. 6, the linearized problem is

$$\begin{aligned}
\min_{\delta\boldsymbol{x}^k} \quad J\left(\delta\boldsymbol{x}^k\right) = {} & \frac{1}{2}\left[\delta\boldsymbol{x}^k + \left(\boldsymbol{x}^{k-1} - \boldsymbol{x}_b\right)\right]^\top \mathbf{B}^{-1} \\
& \left[\delta\boldsymbol{x}^k + \left(\boldsymbol{x}^{k-1} - \boldsymbol{x}_b\right)\right] \\
& + \frac{1}{2}\left(\mathbf{G}^{k-1}\delta\boldsymbol{x}^k - \boldsymbol{d}^{o,k-1}\right)^\top \mathbf{R}^{-1} \\
& \left(\mathbf{G}^{k-1}\delta\boldsymbol{x}^k - \boldsymbol{d}^{o,k-1}\right).
\end{aligned} \tag{7}$$





Each increment, $\delta \boldsymbol{x}^k$, is found in sequential outer loop iterations, where the inner loop solves the quadratic cost function in Eq. 7 using linear optimization. $k$ is the number of the current outer loop. The superscript on $\mathbf{G}^{k-1}$ denotes that it is linearized around the state from the previous iteration, i.e.,

$$
\mathbf{G}^{k-1} = \begin{pmatrix} \mathbf{H}_1|_{\boldsymbol{x}^{k-1}} \\ \vdots \\ \mathbf{H}_i|_{\boldsymbol{x}^{k-1}} \\ \vdots \\ \mathbf{H}_N|_{\boldsymbol{x}^{k-1}} \end{pmatrix} .
\tag{8}
$$

$\boldsymbol{d}^{o,k-1}$ is the innovation between observations and model values in the previous iteration:

$$
\boldsymbol{d}^{o,k-1} = \boldsymbol{y}^o - G\left(\boldsymbol{x}^{k-1}\right) .
\tag{9}
$$

In an emission inversion for a single chemical species, $n = n_x n_y n_t = O\left(10^5 - 10^6\right)$, depending on the domain size and temporal aggregation. Since the number of members in $\mathbf{B}$ is equal to $n^2$, finding its inverse is computationally infeasible. To circumvent that challenge, Barker et al. (2004) implemented a control variable transform (CVT) through a square root

preconditioner, $\mathbf{U}$, in WRFDA. The increment is transformed as $\delta \boldsymbol{x}^k = \mathbf{U} \delta \boldsymbol{v}^k$, where $\mathbf{B} = \mathbf{U}\mathbf{U}^\top$, $\mathbf{U}^\top \mathbf{B}^{-1} \mathbf{U} = \mathbf{I}_n$, and $\mathbf{I}_n \in \mathcal{R}^{n \times n}$ is the identity matrix. The transformed minimization problem is

$$
\begin{aligned}
\min_{\delta \boldsymbol{v}^k} \quad J\left(\delta \boldsymbol{v}^k\right) = {} & \frac{1}{2}\left(\delta \boldsymbol{v}^k - \boldsymbol{d}^{b,k-1}\right)^\top \left(\delta \boldsymbol{v}^k - \boldsymbol{d}^{b,k-1}\right) \\
& + \frac{1}{2}\left(\mathbf{G}^{k-1}\mathbf{U}\delta \boldsymbol{v}^k - \boldsymbol{d}^{o,k-1}\right)^\top \mathbf{R}^{-1} \\
& \left(\mathbf{G}^{k-1}\mathbf{U}\delta \boldsymbol{v}^k - \boldsymbol{d}^{o,k-1}\right) ,
\end{aligned}
\tag{10}
$$

where the background departure, summed over all previous outer iterations, is

$$
\boldsymbol{d}^{b,k-1} = -\sum_{k_o=1}^{k-1} \delta \boldsymbol{v}^{k_o} .
\tag{11}
$$

In addition to circumventing calculating $\mathbf{B}^{-1}$, the preconditioner reduces the condition number of the problem, speeding up the minimization process.

### 2.3.2   Error covariance

WRFDA-Chem utilizes a very similar CVT as WRFDA, with some modification for the scaling factor control variables. The transform $\delta \boldsymbol{x}^k = \mathbf{U} \delta \boldsymbol{v}^k$ is performed through two separate operations as $\mathbf{U} = \mathbf{U}_t \mathbf{U}_h$. Although the horizontal transform ($\mathbf{U}_h$)

only deals with correlations in the $x$ and $y$ directions, and the temporal transform ($\mathbf{U}_t$) only does so in the temporal dimension, they are both $n \times n$, with sub-matrices along the diagonal of dimension $(n_x n_y) \times (n_x n_y)$ and $(n_t) \times (n_t)$, respectively. The computational overhead of multiplying by either transform is reduced by only handling the non-zero elements. $\mathbf{U}_h$ is carried





out using recursive filters (Barker et al., 2004) and the scalar correlation length scale, $L_h$. $\mathbf{U}_t$ is constructed in a similar fashion as the vertical transform in WRFDA (Barker et al., 2004), except that herein we use all of its eigenmodes. The user specifies the duration of emission scaling factor bins (in minutes), the temporal correlation timescale ($L_t$, in hours), and the grid-scale relative emission uncertainty, $\sigma_x$. WRFDA-Chem converts these selections to a covariance sub-matrix $\mathbf{B}_t = \mathbf{\Sigma}\mathbf{C}\mathbf{\Sigma} \in \mathcal{R}^{n_t \times n_t}$,

where $\mathbf{C}$ is the temporal correlation matrix and $\mathbf{\Sigma} = \sigma_x \mathbf{I}_{n_t}$. $\mathbf{B}_t$ is square, symmetric, and positive-definite. Similar to Saide et al. (2015), $\mathbf{C}$ is defined using an exponential decay,

$$C_{ij} = e^{-\frac{\Delta t}{L_t}}, \tag{12}$$

where $\Delta t$ is the time elapsed between the beginning of two particular emission steps. The covariance is decomposed into eigenmodes as $\mathbf{B}_t = \mathbf{E}_t \mathbf{\Lambda}_t \mathbf{E}_t^\top$; these are readily calculated, because the dimension of $\mathbf{B}_t$ is the square of the number of

emission time steps (e.g., 24 steps for hourly scaling factors in a single day inversion). Throughout the optimization, the temporal transform is carried out through multiplication by

$$\mathbf{U}_t = \begin{bmatrix} \mathbf{E}_t \mathbf{\Lambda}_t^{1/2} & \dots & \mathbf{0} \\ \vdots & \ddots & \vdots \\ \mathbf{0} & \dots & \mathbf{E}_t \mathbf{\Lambda}_t^{1/2} \end{bmatrix} \tag{13}$$

and its transpose.

In general, the prior variances are estimated in the form of multiplicative emission uncertainty in $\beta$ space (e.g., "factor

of 2, 3, 4, etc."), not in the exponential CV ($\boldsymbol{x}$) space. The covariances (off-diagonal terms of $\mathbf{B}$) defined previously are assumed to be applicable in CV space. Transformations between the expectations and covariances of a multivariate log-normal ($\boldsymbol{\beta} \sim \mathcal{LN}\left(\boldsymbol{\mu}_{\beta^0}, \mathbf{B}_{\beta^0}\right)$) and a Gaussian distribution (i.e., $\boldsymbol{x} \sim \mathcal{N}\left(\boldsymbol{x}_b, \mathbf{B}_x\right)$) are derived by, e.g., Halliwell (2015), as

$$\mathbb{E}\left[\boldsymbol{\beta}^0\right]_i = \boldsymbol{\mu}_{\beta^0} = \exp\left(x_{b,i} + \frac{1}{2}\mathbf{B}_{x,ii}\right) \tag{14}$$

and

$$\mathbf{B}_{\beta^0,ij} = \exp\left[x_{b,i} + x_{b,j} + \frac{1}{2}\left(\mathbf{B}_{x,ii} + \mathbf{B}_{x,jj}\right)\right]\left(\exp \mathbf{B}_{x,ij} - 1\right), \tag{15}$$

respectively, where $i$ and $j$ are general indices coinciding with individual CV members, $\mathbb{E}$ is the expectation operator and $\exp$ is the natural exponential function. The subscript $\beta^0$ indicates a variable is evaluated in lognormal space in the previous iteration, when $k = 0$, and the subscript $x$ indicates an evaluation in CV space. Since the CVs are normally distributed, $\boldsymbol{x}_b$ is the mean, median, and mode. As Eq. 14 shows, the expected value, or mean, of $\boldsymbol{\beta}^0$ is not equal to its median, $\exp x_{b,i}$, the latter being the

characteristic we use here. The prior linear scaling factor variances are

$$\sigma_{\beta^0,i} = \exp\left[x_{b,i} + \frac{1}{2}(\sigma_{x_b,i})^2\right]\left[\exp(\sigma_{x_b,i})^2 - 1\right]^{\frac{1}{2}}. \tag{16}$$



This is identical to the variance transformation between univariate log-normal and Gaussian distributions. With an initial guess of $\sigma_{x_b,i} = 0$, the recursive inverse relation,

$$\sigma_{x_b,i} = \sqrt{\log\left[1 + \frac{\left(\sigma_{\beta^0,i}\right)^2}{\left(\mu_{\beta^0,i}\right)^2}\right]}$$

$$= \sqrt{\log\left[1 + \frac{\left(\sigma_{\beta^0,i}\right)^2}{\exp\left(2x_{b,i} + \left(\sigma_{x_b,i}\right)^2\right)}\right]}. \tag{17}$$

converges for reasonable ranges of $\sigma_{\beta^0,i}$, which is the additive uncertainty in $\beta$. Earlier emission inversion works (e.g., Elbern
et al., 2007) assume that

$$\left(\sigma_{\beta^0,i} + 1\right)^2 \approx \frac{\exp\left(x_{b,i} + \sigma_{x_b,i}\right)}{\exp\left(x_{b,i} - \sigma_{x_b,i}\right)} = \left(\exp\sigma_{x_b,i}\right)^2,$$

which is equivalent to

$$\sigma_{\beta^0,i} + 1 \approx \exp\sigma_{x_b,i}, \tag{18}$$

and its inverse

$$\sigma_{x_b,i} \approx \log\left(\sigma_{\beta^0,i} + 1\right). \tag{19}$$

$(\sigma_{\beta^0} + 1)$ is the multiplicative error in emissions. For example, $\sigma_{\beta^0} = 2$ gives a factor of three ($\times 3$) uncertainty. In our case $x_{b,i} = 0$ and Eq. 18 gives an error in $\sigma_{\beta^0,i} + 1$ less than 3% for $\sigma_{x_b} \in [0, \log(2)]$, but reaches 100% mismatch at $\sigma_{x_b} = \log(4.2)$. Previous works that use Eq. 19 with relative emission errors less than $\times 3$ do not warrant corrections. However, utilizing Eq. 17 is important for high-resolution inversions of BB sources, where grid-scale uncertainties are probably above that threshold.
Sections 3.3 and 4 include further discussion of emission uncertainty.

The observation-model covariance matrix, $\mathbf{R}$, is assumed diagonal. For each $p$ measurement, the total variance is defined as the sum of observation $\left(\sigma_{p,o}^2\right)$ and model $\left(\sigma_{p,m}^2\right)$ components, following the approach by Guerrette and Henze (2015). $\sigma_{p,m}$ is determined from an ensemble of 156 WRF-Chem model configurations. Each member uses a unique combination of options for PBL mixing, surface layer, LSM, and longwave and shortwave radiation options, as well as includes or excludes
microphysics and subgrid cumulus convection. $\sigma_{p,o}$ accounts for instrument precision, representativeness error, and averaging of measurements to the model resolution. We do not use the weighting term previously defined by Guerrette and Henze (2015), because small residuals with low uncertainty do not appear to hinder the inversion process. Refer to that work for more particular details of how $\sigma_{p,m}^2$ and $\sigma_{p,o}^2$ are calculated.





### 2.3.3 Linear optimization

With all of the terms in Eq. 10 defined, the linear optimization proceeds as follows. The inner loop seeks the optimal $\delta\boldsymbol{v}^k$, at which point

$$
\begin{aligned}
\nabla_{\delta\boldsymbol{v}}J &= \left(\delta\boldsymbol{v}^k - \boldsymbol{d}^{b,k-1}\right) \\
&\quad + \mathbf{U}^\top\mathbf{G}^{k-1^\top}\mathbf{R}^{-1}\left(\mathbf{G}^{k-1}\mathbf{U}\delta\boldsymbol{v}^k - \boldsymbol{d}^{o,k-1}\right) \\
&= \mathbf{0}.
\end{aligned}
\tag{20}
$$

The action of $\mathbf{G}^{k-1^\top}$ on a vector is calculated with the ADM. Solving for the CVT increment,

$$
\begin{aligned}
\delta\boldsymbol{v}^k &= \left(\mathbf{I}_n + \mathbf{U}^\top\mathbf{G}^{k-1^\top}\mathbf{R}^{-1}\mathbf{G}^{k-1}\mathbf{U}\right)^{-1}\left(\boldsymbol{d}^{b,k-1} + \mathbf{U}^\top\mathbf{G}^{k-1^\top}\mathbf{R}^{-1}\boldsymbol{d}^{o,k-1}\right) \\
&= -\left[\mathcal{H}_{\delta\boldsymbol{v}}\right]^{-1}\nabla_{\delta\boldsymbol{v}}J|_{\delta\boldsymbol{v}^k=\mathbf{0}},
\end{aligned}
\tag{21}
$$

where $\mathcal{H}_{\delta\boldsymbol{v}} = \nabla^2_{\delta\boldsymbol{v}}J$ is the Hessian of Eq. 10. $\mathcal{H}_{\delta\boldsymbol{v}}$ and its inverse are too large to store and calculate explicitly. Through an iterative process, the inner loop linear optimization estimates the product of the inverse Hessian with the initial cost function gradient. Finite precision and the problem dimension, $n$, prevent Eq. 20 from being exactly equal to zero. Increasing the number of inner loop iterations to approach such an objective does not necessarily speed up convergence in the nonlinear problem of Eq. 2. Large innovations, $\boldsymbol{d}^{o,k}$, may remain after relinearization around the new state $\boldsymbol{x}^k$.

The two linear optimization algorithms available in WRFDA are Conjugate Gradient and the Lanczos recurrence described on p. 493 of Golub and Van Loan (1996). We use Lanczos, which aids the estimation of posterior error as described in Sec. 2.3.4. Linear optimization strategies are designed to solve a quadratic problem

$$
\begin{aligned}
\min_{\hat{\boldsymbol{x}}} \quad F(\hat{\boldsymbol{x}}) &= \frac{1}{2}\hat{\boldsymbol{x}}^\top\mathbf{A}'\hat{\boldsymbol{x}} - \hat{\boldsymbol{x}}^\top\boldsymbol{b} + c \\
\mathbf{A}'\hat{\boldsymbol{x}} &= \boldsymbol{b}.
\end{aligned}
\tag{22}
$$

The equivalence of incremental 4D-Var (Eqs. 10) and Gauss Newton (GN) to solve Eq. 22 is demonstrated in Appendix A; there, we repeat some derivations by Lawless et al. (2005), Gratton et al. (2007), and Tshimanga et al. (2008) using the notation defined herein. The advantage of this equivalence is that any studies pertaining to issues and advances with GN have the potential to inform incremental 4D-Var; we exploit this in Sec. 2.3.5 to improve the relinearization behavior for nonlinear CVs.

### 2.3.4 Posterior Error

Posterior uncertainty is a useful measure to diagnose the value of an emission inversion. In a region of linear behavior of the full cost function, Eq. 2, and when $\delta\boldsymbol{x}$ is normally distributed, the posterior covariance, $\mathbf{A}$, is equal to the inverse Hessian of Eq. 7 (e.g., Thacker, 1989; Fisher and Courtier, 1995):

$$
\mathbf{A} = \left[\mathcal{H}_{\delta\boldsymbol{x}}\right]^{-1},
\tag{23}
$$



where

$$\mathcal{H}_{\delta\boldsymbol{x}} = \mathbf{B}^{-1} + \mathbf{G}^{k-1^\top}\mathbf{R}^{-1}\mathbf{G}^{k-1}. \tag{24}$$

Combining this with the expression for the Hessian of Eq. 10 we used in Eq. 21 gives a conversion from the transformed variable space

$$\mathcal{H}_{\delta\boldsymbol{v}} = \mathbf{U}^\top \mathcal{H}_{\delta\boldsymbol{x}}\mathbf{U}. \tag{25}$$

Using a Lanczos recurrence to solve the inner loop optimization problem in Eq. 10 has the benefit of producing the means to approximate $[\mathcal{H}_{\delta\boldsymbol{v}}]^{-1}$, which we demonstrate in Appendix A. The final result of that derivation is the posterior error,

$$\mathbf{A} = \mathbf{U}[\mathcal{H}_{\delta\boldsymbol{v}}]^{-1}\mathbf{U}^\top$$
$$\approx \mathbf{B} + \sum_{k_i=1}^{l} \left(\lambda_{k_i}^{-1} - 1\right)(\mathbf{U}\hat{\boldsymbol{\nu}}_{k_i})(\mathbf{U}\hat{\boldsymbol{\nu}}_{k_i})^\top, \tag{26}$$

in terms of the eigenvectors of $\mathcal{H}_{\delta\boldsymbol{v}}$, $\hat{\boldsymbol{\nu}}_{k_i} = \mathbf{Q}_l\hat{\boldsymbol{w}}_{lk_i}$. Each inner iteration, $k_i$, leading up to the current iteration $l$ of the Lanczos optimization, produces (1) a new *Lanczos vector* in the orthonormal matrix $\mathbf{Q}_l = [\hat{\boldsymbol{q}}_1,..,\hat{\boldsymbol{q}}_l]$ and (2) a new row and column in a tridiagonal matrix $\mathbf{T}_l$, whose $k_i^{th}$ eigenpair is $(\lambda_{k_i}; \hat{\boldsymbol{w}}_{lk_i})$. $\mathbf{A}$ is a low-rank update to $\mathbf{B}$, because $l << n$ due to the wall-clock requirements of running the TLM and ADM once per iteration. Equation 26 is consistent with earlier publications (Fisher and Courtier, 1995; Meirink et al., 2008).

### 2.3.5 Damped Gauss Newton

Each CV increment, $\delta\boldsymbol{x}^k$, must be small enough to keep the error associated with the tangent linear assumption, Eq. 6, below some threshold. However, the nonlinearity of the log-normal prior emission errors contributes to failures in that respect. For demonstration, we consider the treatment of $\beta \in \boldsymbol{\beta}$ and $x \in \boldsymbol{x}$ associated with a single grid cell. At the end of an outer loop, $x$ is updated, and $\beta$ is relinearized using

$$\beta^k = e^{x^{k-1} + \delta x^k} = \beta^{k-1}e^{\delta x^k}. \tag{27}$$

Thus, the increment in $\beta$ is

$$\delta\beta^k = \beta^k - \beta^{k-1} = \beta^{k-1}\left(e^{\delta x^k} - 1\right), \tag{28}$$

which reveals the nonlinear nature of the emission increment. This contrasts with the TLM version of the transform in Eq. 5, which states

$$\delta\beta' = e^{x^{k-1}}\delta x = \beta^{k-1}\delta x'. \tag{29}$$

The ratio of $\delta\beta^k/\delta\beta'$ gives the multiplicative error in the tangent linear assumption during relinearization:

$$\epsilon_{TL} = \frac{e^{\delta x^k} - 1}{\delta x^k}. \tag{30}$$





Around $\delta x^k = 0$, the tangent linear relationship very closely matches the nonlinear equation, giving $\epsilon_{TL} \approx 1$. For $\delta x^k > 0$, $\epsilon_{TL}$ grows nearly exponentially toward $\infty$, reaching $\times 2$ at $\delta x^k \approx 1.26$. When $\delta x^k < 0$, $\epsilon_{TL}$ shrinks asymptotically toward zero, reaching $\times 0.5$ at $\delta x^k \approx -1.59$. As $\epsilon_{TL}$ is farther from unity, it is more likely that the linear optimization will generate $J\left(\boldsymbol{x}^k\right) > J\left(\boldsymbol{x}^{k-1}\right)$. Not only do we never want that to happen, but we would prefer to advance toward a more optimal solution
as quickly as possible.

Violation of the TL assumption and potential solutions are discussed in several DA works. The prevailing strategy in chemical 4D-Var is to apply a non-incremental nonlinear optimization strategy (e.g., Henze et al., 2009; Bergamaschi et al., 2009), eliminating the inner-outer loop structure. Implementing this approach in WRFDA with posterior error estimation would be a considerable additional effort. The use of the tangent linear model in the inner loop also presents computational advantages for
dual resolution multi-incremental 4D-Var (e.g., Zhang et al., 2014b). Alternatively, Gratton et al. (2013) discuss application of GN in a trust region framework, which has the limitation that a portion of the computationally expensive outer loop increments will be rejected. Some authors have successfully applied the Levenberg-Marquardt algorithm in EnKF (e.g., Chen and Oliver, 2013; Mandel et al., 2016) by adding a regularization term to the cost function.

A simpler approach yet is damped GN (DGN), which changes the inner loop increment in Eq. 21 to

$$\delta \boldsymbol{v}^k = -\eta^k \left[\mathcal{H}_{\delta \boldsymbol{v}}\right]^{-1} \nabla_{\delta \boldsymbol{v}} J|_{\delta \boldsymbol{v}^k = \boldsymbol{0}}, \qquad (31)$$

and uses a line search to find an optimal scalar $\eta^k \in (0,1]$ at the completion of each outer loop iteration (Kelley, 1999). DGN is based on the Armijo rule, which states that the increment found by GN points toward a direction of lower $J$; if the step size terminus is outside the linear behavior of the model, decrease the step size. WRFDA-Chem uses a non-optimal variant we call heuristic DGN, that requires user intervention to determine $\eta^k$. Results with a simplified test problem in MATLAB indicate that
the resultant CV's near the optimum are nearly identical either with the line search or heuristic damping. However, heuristic DGN likely increases the number of outer iterations required to converge, and motivates implementing the line search in future work. The same MATLAB tests showed that applying a range of damping coefficient values before the Lanczos process has no impact on the estimated $\mathcal{H}_{\delta \boldsymbol{v}}^{-1}$.

The heuristics to determine $\eta^k$ are a function of the prior covariance. As the uncertainty increases, $\eta^k$ should be smaller,
because the initial gradient and resulting increments will be larger in magnitude. Additionally, $\eta^k$ should increase in each subsequent outer loop iteration as the nonlinear optimum is approached, since the diminishing increment magnitude will eventually satisfy the tangent linear assumption. We found that a prior multiplicative emission uncertainty of $\times 3.8$, coinciding with CV uncertainty of $\sigma_x = 1.099$, requires $\eta^0 = 0.4$ in the first outer loop iteration. $\eta^0$ should be adjusted in inverse proportion to $\sigma_x$. Presumably there is some lower limit of $\sigma_x$ where no damping is required. In WRFDA-Chem, the damping ramps linearly
back to 1 in the final outer loop.



## 3   ARCTAS-CARB Case Study

### 3.1   Inversion setup

From late May until 20 June 2008, the southwest U.S. experienced a very dry period with little to no cloud cover appearing in MODIS true color imagery, and no recorded rainfall for most of California. On 21 June, the Aqua and Terra satellites recorded cloud cover for much of Northern California, south of San Francisco, and along the Sierra Nevada mountain range, and there were wide-spread lightning strikes over night. As is shown in Fig. 1, there was a spike in fire detections during the night between 21 and 22 June. Thus, from the morning through evening of 22 June, California experienced a transient fire initiation event. The wild fires burned well into July, exacerbating poor air quality throughout the state. The 22 June flight of ARCTAS-CARB disembarked from Los Angeles, swept out over the ocean, flew directly through smoke from forest fires in Northern California, then returned down the coastline. That flight encountered anthropogenic sources of BC in the morning, and BB sources for the remainder after returning to land. The 24 June flight passed back and forth in the downwind region between Los Angeles and San Diego, measuring the outflow from those cities and the transportation lines between them. A third flight on 26 June flew in the free troposphere from Los Angeles, north over the fires, and exited the model domain to the east.

We use WRFDA-Chem 4D-Var to constrain BB and anthropogenic aerosols on three days during ARCTAS-CARB using aircraft and IMPROVE surface observations. We utilize aircraft measurements of absorbing carbonaceous aerosol at $10\,s$ intervals from the single particle soot photometer (SP2) on 22, 24, and 26 June (Sahu et al., 2012). For this study, we assume equivalency between the SP2 measurement and modeled BC, and re-average to the $90\,s$ model time step using the revision 3 product, a process described in Guerrette and Henze (2015). We also use 24-hour average surface observations of light absorbing carbon (LAC) on 23 and 26 June (Malm et al., 1994), assuming an equivalence with modeled BC, and ignoring the 7% high bias relative to the SP2 found by Yelverton et al. (2014). All treatments of observations are identical to those described in Guerrette and Henze (2015), including an analysis of model-observation BC mismatch that feeds into the inverse modeling study.

Using measurements from 22, 23, and 24 June, the 4D-Var system constrains anthropogenic and BB sources simultaneously. Data collected between 07:00:00-16:00:00 LT on 22 June is used in an inversion from 22 June, 00:00:00 UTC to 23 June, 00:00:00 UTC, during which time WRF-Chem is run freely, without nudging. The emission scaling factors for this 24-hour time period for both source types are applied to subsequent days from 23-26 June in a cross validation experiment. The 24 and 26 June aircraft and 23 and 26 June surface observations are used to analyze the utility of observationally constrained scaling factors found on one day to fix source errors on subsequent days. The 23 and 24 June surface and aircraft data is used in a 48-hour inversion from 23 June, 00:00:00 UTC to 25 June, 00:00:00 UTC, also without nudging. Cross validation is performed for these source estimates using 26 June surface and aircraft data.

Through preliminary testing, we found that horizontal correlation length scales on the order of the grid spacing provides the lowest posterior cost function. For both time periods, this length scale is set to twice the grid scale, $L_h = 36\,\mathrm{km}$. The emission scaling factors are aggregated in each hour, which coincides with the emission file reading interval for both source



types. The correlation scale is set to $L_t = 4\,\mathrm{h}$, following Saide et al. (2015). In addition to spreading error information across adjacent grid cells, the correlation scales reduce the effective number of CVs. Through sensitivity tests where we considered the smoothness of the posterior and the stationary posterior cost function value, and after consulting published values for regional emission uncertainties (see Sec. 1) in different global settings, we use a grid-scale BB uncertainty of $\times 3.8$. The BB uncertainty might also be approximated from the ratio of prior domain-wide total emissions between FINNv1.0 and QFED, which is given in Table 5 as $\times 3.5$. If the median emission strength lies in the middle of QFED and FINNv1.0, then the prior domain-wide relative uncertainty is $\times\sqrt{3.5} = \times 1.8$. The uncertainty would then need to be inflated further to account for spatial and temporal disaggregation and the possibility that grid-scale sources from the two inventories do not bound the true value. The prior anthropogenic grid-scale uncertainty is set to $\times 2$, which is within the reasonable bounds discussed in Sec. 1.

In addition to these standard settings, several sensitivity scenarios are used to gauge the sensitivity of the posteriors during two time periods to alternative inversion settings. The full set of scenarios are summarized in Table 1, and are as follows. FINNv1.0 is used as the default BB inventory in a scenario called FINN_STD for both inversion periods. QFED_STD uses the QFEDv2.4r8 BB inventory. Both FINN_L18 and QFED_L18 use $L_h = 18\,\mathrm{km}$. FINN_V1.5 utilizes the FINNv1.5 BB inventory. For the 23/24 June inversion, we show results for both QFED_STD and FINN_STD, the latter of which includes variations where either surface or aircraft observations are excluded. The number of aircraft observations is $N_{\mathrm{obs}} = 241$ on 22 June and $N_{\mathrm{obs}} = 302$ on 24 June. There were $N_{\mathrm{obs}} = 35$ active surface sites on 23 June, 13 of them within California. We use six outer iterations consisting of 10 inner iterations each. Given the number of inner iterations used, and the wall-time of the tangent linear plus the adjoint ($10\times$ the nonlinear model), the cost of incremental 4D-Var is approximately $600\times$ that of a single forward simulation, which is much cheaper than using finite difference methods to approximate derivatives instead of the linearized models when $n \sim 10^5$.

### 3.2 Posterior model performance

The convergence properties of the 22 and 23/24 June inversion scenarios are shown in the outer loop cost function progression in Fig. 3. All of the 22 June scenarios led to comparable cost function values at numerical convergence, as shown in Fig. 3. The gradient norms are also reduced by nearly two orders of magnitude in all cases. The $\chi^2$ criteria states that the posterior cost function should be equal to $\frac{1}{2}N_{\mathrm{OBS}}$. In all of the scenarios, $J$ converges to approximately $N_{\mathrm{OBS}}$, indicating that a portion of the model errors are not fully spanned by prior emission errors. For the 23/24 June inversion, QFED_STD reaches a lower cost function value, and both scenarios achieve similar $\chi^2$ values as the 22 June cases. Scrutinizing other sources of error (e.g., initial and boundary conditions for BC and meteorological variables, transport, BB plumerise, and model discretization) either independent from source strengths or simultaneously in the inversion framework should elicit further cost function reductions.

The non-emission sources of error for 22 June are evident in the time series in Fig. 4. The posterior is within the combined model/observation uncertainty (see Sec. 2.3.2) much more often than the prior. The only time during the inversion when the forecast degrades is for an observed peak at 22 June, 08:00 LT. Model uncertainty is higher in locations where the prior concentration is higher, due to variability in the configuration ensemble boundary layer heights (Guerrette and Henze, 2015). This high prior uncertainty in $\mathbf{R}$ allows the stronger constraint at 08:30 LT to dominate the morning anthropogenic emissions,




since this flight portion was confined to Los Angeles. In the afternoon, when the DC-8 passed over the wildfires, an increase in posterior emissions captures several of the observed BC peaks. The posterior is able to match the high-resolution variability of the observations at 13:30 LT, which may support the validity of the temporal averaging scheme.

The $R^2$ coefficients and slopes for linear fits between the prior and posterior and both aircraft and surface observations

are summarized in Tables 2 and 3. Those results include cross-validation data on non-inversion days, which is discussed in Sec. 3.5. For both inversion periods, there are considerable model performance improvements for observations that are used in the inversion. FINN_STD improves $R^2$ from 0.11 to 0.82 and slope from 0.26 to 0.8 on 22 June. QFED_STD improves $R^2$ from 0.03 to 0.73 and slope from 0.34 to 0.71. Similar improvements occur for 23 June surface observations during the 23/24 June inversion. The posterior match to 24 June aircraft observations is improved, but not nearly as much as the other two data

sets. The 22 June inversion results are also shown in the first row of Fig. 6, where the progression of the fit parameters is shown for the multiple scenarios. While all scenarios show similar improvements, the FINN_STD and QFED_STD results indicate the posteriors are still underpredicting many low and high concentrations. Overprediction seems to be less of a problem. A similar phenomenon occurs for the 24 June observations in Fig. 6 in the inversion that uses both surface and aircraft observations. On both 22 and 24 June, the remaining low bias is either due to large prior observation and model error (diagonal of **R**) or due to

the prior errors not being sensitive to emission increments.

### 3.3 Posterior emissions

Figure 7 shows the prior and posterior BB emissions for FINN_STD and QFED_STD during both simulation periods. In that figure there are several outlined emission areas (EAs); each EA was chosen to identify regions where a subset of the grid-scale analysis increment ($\delta\boldsymbol{x}_{\mathrm{EAX}} \subset \delta\boldsymbol{x}$) from both prior inventories is of similar sign. The coordinates of the EAs are listed in Table 4.

The two inversions do not reach identical total posterior BC emissions, but they do converge in certain aspects. Table 5 gives the emission subtotals for the EAs. During both inversions, each EA has emission increments of the same sign for both scenarios. Therefore, while domain-wide sources seem to be bounded by the two priors (as evidenced by their convergence), the same might not be true within the individual EAs. EA3, which accounts for the smallest average posterior total, is the only region where the magnitude of the log-ratio between QFED and FINN is smaller in the posterior on 22 June. The ratio is reduced in

EA2, but there the FINN posterior is $\times 2$ larger than that for QFED. On 23/24 June, the two scenarios have less posterior spread in all of the EAs. Although Table 5 indicates large changes in source strengths across the EAs, Figure 8 reveals that a majority of the absolute emission increment (posterior minus prior) in both FINN_STD and QFED_STD arose in only a few grid cells, often where the prior has the largest magnitude. The linear scaling factor pattern is similar between the two scenarios, with those for QFED_STD shifted toward decreases due to the high prior bias.

The temporal distribution of prior and posterior BB emissions within the four EAs are shown in Fig. 9 across all inversion scenarios on 22 June. The FINNv1.5 prior is an extreme outlier on the local afternoon of 21 June for EA1, EA2, and EA4. The same is true all day on 22 June for EA2, where the posteriors from other scenarios adjust toward the FINNv1.5 prior. Meanwhile, in other times when FINNv1.5 appears to converge toward the posteriors found using the other two priors, the prior uncertainty of $\times 3.8$ is too restrictive to allow full convergence, since the priors differ by $\times 10$. EA1 is characterized by





decreases for all scenarios at all times. EA2, EA3, and EA4 exhibit early morning peaks between 03:00 and 06:00 LT that were not captured in the prior. In separate sensitivity tests, these peaks only appear when $L_t > 1\,\mathrm{h}$, and become more prevalent as $L_t$ is increased. Saide et al. (2015) attributed similar behavior in posterior estimates of the 2013 Rim Fire to persistent large scale burning. Zhang et al. (2012) found similar, less pronounced bimodal behavior for all of North America, which could be

more noticeable in a regional inversion. Another possibility on 22 June 2008 is that the early morning burning is caused by the transient fire initiation event, which would explain the ramping of emissions for the QFED and FINNv1.5 posteriors in EA2. For both QFED and FINNv1.0, reducing the correlation length to $L_h = 18\,\mathrm{km}$ reduces the analysis increment in all EAs. This is especially apparent in EA4 for FINN_L18, where the increment is negligible.

The differing diurnal patterns in EA2 across scenarios could be attributed to variation in plume heights, QFED regridding

errors, and the regularization term of the cost function. The observations most sensitive to EA2 sources were captured within or very near fire plumes. Plume heights are calculated hourly in an online 1D vertical mixing scheme in WRF-Chem (Freitas et al., 2007, 2010; Grell et al., 2011), which depends strongly on burned areas. With FINN, the areas are provided for each fire independently, while for QFED the areas use a default value of $0.25\,\mathrm{km}^2$ per fire. In both cases, the maximum area burned per grid cell per day is $2\,\mathrm{km}^2$. The regridding error discussed in Sec. 2.2 introduces fire locational errors, especially in EA2.

A small error in vertical or horizontal mapping of a discrete point source on the model grid could hinder the optimization in distinguishing it from others. The uniform relative uncertainty in the prior inhibits consolidation of multiple posteriors when the prior spread is heterogeneous and sometimes very large. Quantifying the heterogeneity of uncertainty could contribute to posterior agreement between inversions using different priors, as well as to reducing the cost function.

The spread of local emissions provide some sense of that heterogeneity. Each EA covers a region approximately the size of

a grid box in a global simulation with a chemical transport model. Due to the nature of variance aggregation, uncertainty grows as the grid scale gets smaller. In individual EAs, the spread between FINNv1.0 and QFED priors is $\times 2$-$\times 6$ for both hourly (Fig. 9) and daily (Table 5) strength on 22 June. If the median emission strength lies in the middle, then a proxy for prior EA relative uncertainty is $\times\sqrt{2} - \times\sqrt{6} = \times 1.4 - \times 2.4$. Since the two inventories use identical diurnal patterns, the hourly estimate is missing information about uncertainties in daily emission timing. Using the posterior spread in a similar way gives

approximate EA uncertainties of $\times\sqrt{3} - \times\sqrt{10} = \times 1.7 - \times 3.2$ on hourly scales and $\times\sqrt{2} - \times\sqrt{7} = \times 1.4 - \times 2.6$ on daily scales. This posterior estimate accounts for contributions in the prior definitions, including regridding, plume rise, and diurnal patterns. These ranges provide much more detail estimates than simply taking the domain-wide ratio of total emissions for the campaign period. However, the spread is itself missing information about uncertainty that could be found through carrying out similar inversions across an ensemble of model configurations and meteorological initial and boundary conditions (e.g.,

Lauvaux et al., 2016), or by comparing many more inventory priors (e.g., Zhang et al., 2012) and posteriors. All this is to say that the BB inventories used in this study are not provided with analytical estimates of uncertainty, and a lack of information for deriving such values at hourly grid-scales is a topic for future research.

Figure 10 shows the total prior and posterior anthropogenic emissions and Fig. 11 displays the analysis increment and linear scaling factor for FINN_STD on 22 June and separately on 23-24 June. The only difference in QFED_STD, not shown here, is





that anthropogenic scaling factors are shifted in the negative direction in the posterior, likely due to the higher bias in that BB prior. The increments found in a new set of EAs are presented in Table 6.

The 23 and 24 June observations provide much more detailed information about anthropogenic sources. The analysis increment reveals potentially misrepresented city-level emissions in the NEI05 prior. Posterior BC near Barstow, Victorville/Hesperia, Fresno, Edwards Air Force Base, and El Centro/Calexico are increased, while sources near the three coastal cities are decreased. Since Barstow is a crossroads for the BNSF and the Union Pacific railroads, and since Fresno, Victorville/Hesperia, and El Centro/Calexico lie at switching locations for major rail lines, it could be speculated that the prior is missing diesel rail sources of BC. Another possibility is that low bias fire emissions north of Fresno are responsible for the prior underpredictions of 23 June surface concentration measurements exceeding $2\,\mu\mathrm{g\,m}^{-3}$ (see Fig. 5 of Guerrette and Henze, 2015). This is corroborated by the posterior BB emissions being scaled up near Fresno on 23 and 24 June, and by the much smaller model bias for IMPROVE on 22 June before the fires started.

There are also small negative increments near Los Angeles (EA6) and San Francisco (EA7) during both the 22 June and 23/24 June inversions, which are likely attributable to on-road mobile sources. These results are consistent with model bias in surface and aircraft observations on 20 June near both of those cities (Guerrette and Henze, 2015). McDonald et al. (2015) found a decreasing trend in ambient measurements of BC and in a fuel-based bottom-up inventory for both Los Angeles and San Francisco from 1990 to 2010 that might not be captured for the 2008 model year by the snapshot in NEI05. Using a similar fuel-based approach, Kim et al. (2016) derived 2010 CO emissions in the South Coast Air Basin surrounding Los Angeles that are $\times\frac{1}{2}$ the magnitude of NEI05. On-road and other mobile sources make up 36% and 62% of that difference, respectively, and their bottom-up inventory matches more closely with NEI 2011. While not a perfect comparison to BC in 2008, the sign of error in NEI05 relative to the coastal posterior and that study is consistent. An inventory with sector-specific break downs of BC emissions, and additional inversions with more thorough speciated local observations, and higher resolution would all be required to investigate sector-specific anthropogenic pollution.

### 3.4 Error diagnostics

Analysis of posterior emissions uncertainties is useful for understanding the value of the posterior emissions themselves. The diagonal terms of $\mathbf{A}$ are the posterior variances, $\boldsymbol{\sigma}_{x_a}$, which are always smaller than prior variances. The variance reduction could instead be presented in $\beta$ space, by utilizing Eq. 16. However, $\sigma_{\beta^k,i}{}^2 < \sigma_{\beta^0,i}{}^2$ is not guaranteed when $x_{a,i} > x_{b,i} = 0$, because the posterior relative emission uncertainty depends on $x_{a,i}$. For this work, the reductions in variance are presented in CV space. The low-rank estimate of $\mathbf{A}$ is only valid for linear perturbations away from $\boldsymbol{x}_a$. The final outer loop estimate of $\mathbf{A}$ is the most accurate, since it is linearized around the state preceeding $\boldsymbol{x}_a$. A quantitative measure of error reduction in the $k_o^{\mathrm{th}}$ outer loop in the $i^{\mathrm{th}}$ CV is

$$\rho_{i,k_o} = 1 - \left(\frac{\mathbf{A}_{i,i}}{\mathbf{B}_{i,i}}\right)_{k_o} \in [0,1). \tag{32}$$

Values of $\rho_{i,k_o}$ closer to 1 reflect locations where the observations provide a stronger constraint than the prior. This estimate may not reflect the entire error reduction, since it does not capture potential reductions in previous outer loops. Without propagating





updated estimates for $\mathbf{B}$ to subsequent outer loops (e.g., Tshimanga et al., 2008), we also define $\rho_{\mathrm{agg}}$, a qualitative metric that accounts for increases in curvature (decreases in error) in all outer loops:

$$\rho_{i,\mathrm{agg}} = 1 - \prod_{k_o=1}^{k} \left( \frac{\mathbf{A}_{i,i}}{\mathbf{B}_{i,i}} \right)_{k_o} \in [0, 1). \tag{33}$$

$\rho_{i,\mathrm{agg}}$ reveals additional information about observation footprints not shown by $\rho_{i,k_o=6}$. The nonlinear nature of the problem

means $\rho_{i,\mathrm{agg}}$ is not quantitative.

Both ($\rho_{k_o=6}$) and $\rho_{\mathrm{agg}}$ are presented in Figs. 12 and 13 for the BB and anthropogenic members of $\boldsymbol{x_a}$, respectively. 50 inner loop iterations were taken in the final outer loop to improve $\rho$ estimates. $\rho_{k_o=6}$ is $< 45\%$ across all scenarios, except for QFED_STD BB sources near the IMPROVE sites on 23/24 June. If the inner loop were halted at 10 iterations, the error reduction estimates are reduced by up to $\sim 10\%$ (i.e., 35% instead of 45%) in the darkest grid cells. Further decreasing uncertainty

would require observing the same phenomena more thoroughly, either for longer periods, with greater spatial coverage, or with more instruments. The BB error reduction shown in Figure 12 has similar spatial distributions for FINN_STD and QFED_STD scenarios, but differs significantly between the two time periods due to the different spatial coverage of the observations. The reductions in the north on 22 June are more disperse for QFED_STD, which could be caused by the same regridding errors and plumerise differences that influence the posterior emissions. There is also more error reduction in the south for the QFED_STD

emissions. In general, the grid-scale uncertainty improvement is confined to sources close to the observations.

The most obvious application of $\rho$ is to evaluate the footprint of a set of measurements. For example, the large relative BB emission increments in EA1-EA3 on 23/24 June indicate that distant observations can have a large impact on the posterior emissions magnitudes. However, $\rho_{i,k_o=6}$ in Fig. 12 indicates there is nearly zero uncertainty reduction for those emissions. Also, upon considering the last two columns of Table 6, one might conclude that there is a missing weekend (22 June) to

weekday (23/24 June) variation in BC emissions within EA8-10. However, Fig. 13 shows that the 22 June observations only weakly reduce uncertainty in emissions.

In a more tangible application, $\rho$ can be used to assess existing and and future observing strategies in a similar way to how Yang et al. (2014) used adjoint sensitivity information to plan future meteorological observing sites to improve forecasts of extreme dust events in the Korean peninsula. Fig. 13 presents anthropogenic $\rho$ for different combinations of surface and

aircraft observations on 23/24 June. The surface observations primarily resolve sources near Fresno, and to a lesser extent near Los Angeles. Since the purpose of the IMPROVE network is to measure background concentrations, it is mostly successful on 23 June in not being influenced by anthropogenic sources of BC from the major cities. If the goal were to measure anthropogenic sources, inflows, or domain-wide concentrations on daily time scales, then $\rho$ would suggest using a different surface network distribution. Such a conclusion does not conflict with the success of using IMPROVE observations to provide

top-down constraints on both BB and anthropogenic emissions on monthly time scales (e.g., Mao et al., 2015).

Another piece of information useful for comparing observing configurations and inversion scenarios is the trace of the resolution matrix, or degrees of freedom for signal, i.e.,

$$\mathrm{DOF} = \mathrm{Tr}\left[ \mathbf{I}_n - \mathbf{A}\mathbf{B}^{-1} \right], \tag{34}$$



which is equal to the number of modes of variability in the emissions that are resolved by the observations (Wahba, 1985; Purser and Huang, 1993; Rodgers, 1996). Substituting the approximation for $\mathbf{A}$ from Eq. 26,

$$
\begin{aligned}
\mathrm{DOF} &\approx n - \mathrm{Tr}\left[\left(\mathbf{B} + \mathbf{U}\left(\sum_{k_i=1}^{l}\left(\lambda_{k_i}^{-1} - 1\right)\hat{\boldsymbol{\nu}}_{k_i}\hat{\boldsymbol{\nu}}_{k_i}^{\top}\right)\mathbf{U}^{\top}\right)\mathbf{B}^{-1}\right] \\
&\approx -\mathrm{Tr}\left[\mathbf{U}\left(\sum_{k_i=1}^{l}\left(\lambda_{k_i}^{-1} - 1\right)\hat{\boldsymbol{\nu}}_{k_i}\hat{\boldsymbol{\nu}}_{k_i}^{\top}\right)\mathbf{U}^{\top}\mathbf{B}^{-1}\right].
\end{aligned}
\tag{35}
$$

Since $\mathbf{U}$ is square, $\mathbf{U}^{\top}\mathbf{B}^{-1}\mathbf{U} = \mathbf{I}_n$, and $\mathrm{Tr}\left[\hat{\boldsymbol{\nu}}_{k_i}\hat{\boldsymbol{\nu}}_{k_i}^{\top}\right] = \hat{\boldsymbol{\nu}}_{k_i}^{\top}\hat{\boldsymbol{\nu}}_{k_i} = 1$, the expression simplifies as

$$
\begin{aligned}
\mathrm{DOF} &= -\mathrm{Tr}\left[\left(\sum_{k_i=1}^{l}\left(\lambda_{k_i}^{-1} - 1\right)\hat{\boldsymbol{\nu}}_{k_i}\hat{\boldsymbol{\nu}}_{k_i}^{\top}\right)\mathbf{U}^{\top}\mathbf{B}^{-1}\mathbf{U}\right] \\
&\approx \sum_{k_i=1}^{l}\left(1 - \lambda_{k_i}^{-1}\right)\mathrm{Tr}\left[\hat{\boldsymbol{\nu}}_{k_i}\hat{\boldsymbol{\nu}}_{k_i}^{\top}\right] \\
&\approx \sum_{k_i=1}^{l}\left(1 - \lambda_{k_i}^{-1}\right).
\end{aligned}
\tag{36}
$$

Therefore, the only information needed to compute DOF are the eigenvalues of $\mathbf{T}_l$. Each inner loop, $k_i$, has the potential for constraining one additional mode of variability in the emission scaling factors. For all of our inversion scenarios, the leading eigenvalue is on the order of $10^2 - 10^3$, which is equal to the condition number of the full-rank Hessian. As the Lanczos optimization proceeds, each subsequent $\lambda_{k_i}$ is smaller, asymptotically approaching unity, and each eigenmode provides less information than the one preceding it about scaling factor variability.

Figure 14 gives three estimates of DOF at each level of truncation in the final outer loop, that is if higher degrees of eigenvalues were ignored. In that figure, we plot eigenvalue spectra of the FINN_STD and QFED_STD scenarios on 22 June. Similar to $\rho$, we use a 50 iteration linear optimization to improve the bounds on DOF. The $k_i = l$ estimate of the eigenvalue spectrum at each iteration is represented by a single colored line. Each member of the eigenvalue spectrum, represented by vertical grid lines in Fig. 14, converges toward an upper bound as more iterations are taken. Initial guesses for the least dominant eigenvalues are less than 1 for $k_i \geq 8$ for FINN_STD, but they exceed 1 after an additional iteration, consistent with the properties of the Lanczos sequence. The first DOF in parentheses adheres to the philosophy that only converged eigenvalues should be used to estimate DOF; it excludes $\lambda_{k_i}, \ldots, \lambda_l$ such that $\lambda_{k_i}$ is more than 5% changed from the previous estimate. The second DOF in parentheses uses all of the current estimates of the eigenvalues available in iteration $l$. This is still a conservative estimate of DOF, because the true eigenvalues of the full-rank $\mathbf{T}_n$ are always larger than their current numerical estimate. After enough iterations, the numerical growth in DOF is very small, and further computation is not warranted. As the eigenvalue spectra in Figure 14 and the cost function reduction in Figure 3 show, this is long after the cost function is converged enough for practical purposes. The posterior CVs, which are the primary result from inverse modeling, do not change significantly in the final outer loop. Finally, the best estimates of DOF in red brackets are evaluated at different truncations using the most-converged values of the eigenvalues found in the 22nd iteration.





Similar to $\rho$, the quantitative application of DOF is limited to the final outer loop, when $\delta x^n$ is small enough that $(\mathcal{H}_{\delta v})^{-1}|_{x^{n-1}} \approx$ $(\mathcal{H}_{\delta v})^{-1}|_{x^n}$. Absent the need to estimate the posterior Hessian, the outer loop could be ended an iteration earlier. In the inner loop, truncated estimates of $\mathcal{H}^{-1}$ and its eigenvalue spectrum at earlier iterations will provide conservative values for both DOF and $\rho$. The actual DOF is higher than any value shown in Figs. 14 (22 June) and 15 (23/24 June). Therefore, the 22 June observations constrain >14 modes of hourly grid-scale variability through 4D-Var in both the FINN_STD and QFED_STD scenarios. Just like for $\rho$, the optimization constrains additional modes in the earlier outer loop iterations, but that quantification is not straightforward since DOF is defined for linear behavior. If all outer loops were similar, then the total DOF for the entire nonlinear optimization is on the order of 30 to 40.

As shown in Fig. 15, the DOF on 23 and 24 June after 50 iterations are 10, 17, and 23 for the SURF, ACFT, and FINN_STD(23/24) scenarios, respectively. The relative magnitudes show that using combined surface and aircraft observations provides an additional value over using either independently, although the two platforms might have some redundancy. This conclusion is consistent with the maps of BB and anthropogenic $\rho$ in Fig. 13, where the footprints of SURF and ACFT have slight overlap near Los Angeles, but are otherwise independent. Additionally, the higher DOF of ACFT is consistent with its more widespread and larger magnitude $\rho$ values. The slower eigenvalue convergence when both observing types are utilized means that additional inner iterations could yield higher estimates for DOF in that case. What is even more clear, and intuitive, is that $\rho$ and DOF estimates require more iterations as the number of constrained CVs increases, which is directly dependent on the number of observations. What might be of particular interest to future measurement planning is that daily average surface data can provide a useful constraint when captured near sources on the same day as the emission event, which is supported by the sparse $\rho$ map for SURF in Fig. 13, and the large spike near Fresno.

## 3.5 Cross Validation

As an additional evaluation of the robustness of the emission scaling factors, we apply them in cross validation tests. In two separate evaluations, the 22 June scaling factors are applied to 23-26 June emissions, and the 23/24 scaling factors are applied to 25-26 June emissions. Even before carrying out such a test, the heterogeneous adjoint sensitivity signs and magnitudes for each source sector we found on each day of the campaign (Guerrette and Henze, 2015) are an indication that corrective scaling factors in each day will be unique. In that work, we found that the 24 June observations were most sensitive to Southern California anthropogenic sources on 24 June and to Northern and Southern California coastal sources of both sectors on 23 June. The 26 June observations were most sensitive to Northern California fires, and the adjoint sensitivities were of opposite sign than on 23 and 24 June.

As shown in Fig. 4, the cross validated 22 June scaling factors rarely generate improvements to model performance, when compared to 24 and 26 June aircraft observations. On 24 June, some of the high bias predictions are corrected, or even overcompensated, but the low bias prior locations are unaffected. Table 2 shows the $R^2$ and slope of the linear trend lines. The scatter of the fit for QFED_STD on 24 June and the slope for FINN_STD on 26 June are slightly improved, but all other metrics degrade. The increase in slope for FINN_STD comes as a result of better fit to very large concentrations above the PBL associated with fire sources on multiple previous days. The posterior scaling factors generated from the 23/24 June inversion



degrade the forecast of aircraft measurements on 26 June. Since the posterior primarily serves to reduce coastal anthropogenic and BB emissions, it is not surprising that it does not improve a low bias prior two days later.

Table 3 includes cross-validated surface measurements on 23 June and 26 June. There is very little change to the modeled surface concentrations as a result of posterior scaling factors in any inversions that only use aircraft observations. Assimilating surface observations on 23 June (Monday) does improve model comparisons to surface observations on 26 June (Thursday). Those small improvements imply that errors are weakly correlated between weekdays. Although it is beyond the information content provided by the observations used in this work, future studies could compare the efficacy of using weak multiday correlation in **B** and the hard constraint of 24 h periodic scaling factors used herein.

Given the differing flight tracks on multiple days, the cross validation results demonstrate the need to repeat observations of similar phenomena. Such a strategy could help eliminate non-emission related sources of uncertainty, and further characterize temporal heterogeneity of inventory errors. Aircraft and surface observations do not appear to be useful for cross-validation of each other over the short timescales and limited set of flights considered here. At least for this study period, when they are not collocated, each provides some unique information to the inversion. Cross-validation might be more successful when using measurements collected over a broader range of prior error behaviors or by considering a less complex problem than California statewide BB emissions.

## 4   Conclusions and future work

We have presented the implementation and an application of incremental chemical 4D-Var using an atmospheric chemistry model with online meteorology in WRFDA-Chem. This work expands on our previous efforts to develop the ADM and TLM in WRFPLUS-Chem (Guerrette and Henze, 2015). This new inversion tool takes advantage of previous developments of meteorological data assimilation in WRFDA (Barker et al., 2005; Huang et al., 2009). That same framework is applied to lognormally distributed emission scaling factors through an exponential transform. We utilize the square root preconditioner for a CVT using horizontal and temporal scaling factor correlations. The Lanczos linear optimization algorithm in the inner loop allows for estimation of posterior error and DOF for objectively evaluating observing systems. Outer loop convergence is improved with a heuristic DGN multiplier, which allows the incremental 4D-Var framework to handle the nonlinearity of the lognormal cost function. While the optimizations herein focus exclusively on emissions, which are known to be important drivers of model uncertainty in BC estimates (e.g., Fu et al., 2012; Zhang et al., 2014a), other factors such as meteorology, plume rise and deposition mechanisms may also affect the model's predictions of BC concentrations.

When applied to the ARCTAS-CARB campaign period, it is not clear which prior emissions perform better. If assessment by initial cost function value alone were meaningful, FINNv1.0 performs best. However, that could be due to FINNv1.0 being biased low combined with the assumption of Gaussian distributed model-observation errors. Positive residuals are weighted higher than negative ones, even when relative errors are equal. There could be some improvement to the posterior emissions by implementing the incremental log-normal cost function framework derived by Fletcher and Jones (2014). If the purpose of the inventory is to provide air quality warnings to the major California cities, then FINNv1.0, FINNv1.5, and QFEDv2.4r8 all have





some built-in high bias that will err on the side of caution. Their inability to reproduce high concentrations near sources either points to a deficiency in the inventories, vertical mixing processes, or the temporal observation averaging procedure followed herein, diagnosis of which would require measurements of plume injection heights and widths. The relative magnitudes of grid-scale fire and anthropogenic emissions make it difficult to simultaneously constrain them without additional information.

More work should be done to improve both bottom-up and top-down estimates of anthropogenic emissions outside of fire events. We also agree with Mao et al. (2015), who recommended multi-species inversions (e.g., BC and CO) to discern specific source sectors.

Through the setup and application of the 4D-Var system, we gained valuable knowledge to guide future modeling and measurement efforts. We found two errors in the diurnal distribution of BB emissions and identified a scaling necessary to

apply QFED to the western U.S. Additionally, the highly heterogenous posterior scaling factors during ARCTAS-CARB raise questions that the limited BB observations during that time period do not answer. (1) Are BB emission errors always heterogeneous, or only during a transient initiation stage like that observed in June 2008? If heterogeneity is consistent outside initiation events, then inversions should apply weaker inter-day correlation than the hard constraint used herein or have independent scaling factors for each day. (2) Are the temporally bimodal posterior emissions realistic, or are they an artifact of the

correlation timescale used? (3) Are the BB plume heights reasonable, and should they follow a diurnal pattern? The current 1D plume rise mechanism in WRF-Chem depends strongly on specified burned areas, which are diurnally invariant and highly uncertain (e.g., Boschetti et al., 2004). The last two questions indicate there is value in continuous night (between 20:00 and 06:00 LT) and day measurements of the same fire region. Since models poorly predict shallow boundary layers, the use of night time observations in 4D-Var would require characterization and subsequent model tuning of those vertical mixing processes.

Furthermore, if it is accepted that high-resolution models are required to accurately predict degraded air quality events, then high spatial and/or temporal resolution concentration measurements from research campaigns or geostationary satellites are necessary to provide the sufficient constraints on inventory errors. The error reduction estimation method provided herein will be useful for planning these future missions.

Future applications of the WRFDA-Chem system developed here may consider improvements such as the following. One

possible way to reduce model uncertainty would be to extend the multi-incremental 4D-Var available in WRFDA (Zhang et al., 2014b) to the new scaling factor CVs. Multi-incremental chemical 4D-Var would use a high-resolution model forecast to generate trajectory checkpoint files (see Guerrette and Henze (2015)), and could take advantage of improvements to chemical transport at higher resolution realized by using online meteorology demonstrated by Grell et al. (2004) and Grell and Baklanov (2011). In addition, FDDA nudging has been shown to improve wind fields and was used successfully in an LPDM emission

inversion (Lauvaux et al., 2016). Even after exhausting methods to improve the posterior, the error contributions from hard-coded descriptions of meteorology can be bounded using ensemble and sensitivity tests (e.g., Angevine et al., 2014; Lauvaux et al., 2016).





## Appendix A: Relating DA and optimization formulations

The linear optimization in the inner loop solves a system

$$\min_{\hat{\boldsymbol{x}}} \quad F(\hat{\boldsymbol{x}}) = \frac{1}{2}\hat{\boldsymbol{x}}^\top \mathbf{A}'\hat{\boldsymbol{x}} - \hat{\boldsymbol{x}}^\top \boldsymbol{b} + c$$

$$\mathbf{A}'\hat{\boldsymbol{x}} = \boldsymbol{b}. \tag{A1}$$

In our case, $\hat{\boldsymbol{x}} \equiv \delta\boldsymbol{v}^k$. The equivalence of Eq. 10 and Eq. A1 is apparent in Tshimanga et al. (2008), who provide a notational

translation between publications on DA and those on minimization algorithms and preconditioners. We repeat their translation

to account for the differences in formulation of Eq. 10 and Eq. 5 in Tshimanga et al. (2008).

The process starts by considering Lawless et al. (2005) and Gratton et al. (2007), who show that incremental 4D-Var is

equivalent to a truncated Gauss-Newton (TGN) optimization algorithm. The incremental 4D-Var cost function is condensed to:

$$\min_{\delta\boldsymbol{v}} \quad J(\delta\boldsymbol{v}) = \frac{1}{2}\mathbf{f}(\delta\boldsymbol{v})^\top \mathbf{f}(\delta\boldsymbol{v}), \tag{A2}$$

where

$$\mathbf{f}(\delta\boldsymbol{v}^k) \equiv \begin{pmatrix} \delta\boldsymbol{v}^k - \boldsymbol{d}^{b,k-1} \\ \mathbf{R}^{-\frac{1}{2}}\left(\mathbf{G}^{k-1}\mathbf{U}\delta\boldsymbol{v}^k - \boldsymbol{d}^{o,k-1}\right) \end{pmatrix}. \tag{A3}$$

This definition of $\mathbf{f}$ is what enables incremental 4D-Var to be characterized as TGN. The remainder of the derivation amounts

to substitutions. GN approximates Newton's method in each quadratic minimization problem, $k$, to solve for the increment $\delta\boldsymbol{v}^k$

in the linearized system

$$\mathcal{H}_{\delta\boldsymbol{v}}\delta\boldsymbol{v}^k = -\nabla J. \tag{A4}$$

This form is equivalent to multiplying Eq. 21 by the Hessian on both sides. In our case, the right-hand side is $\boldsymbol{b} \equiv -\nabla J = -\mathbf{F}^{k-1\top}\mathbf{f}(\delta\boldsymbol{v}^{k-1})$, where

$$\mathbf{f}(\delta\boldsymbol{v}^{k-1}) \equiv -\begin{pmatrix} \boldsymbol{d}^{b,k-1} \\ \mathbf{R}^{-\frac{1}{2}}\boldsymbol{d}^{o,k-1} \end{pmatrix}, \tag{A5}$$

and

$$\mathbf{F}^{k-1} \equiv \nabla_{\delta\boldsymbol{v}^{k-1}}\mathbf{f}\big|_{\delta\boldsymbol{v}^{k-1}} = \begin{pmatrix} \mathbf{I}_n \\ \mathbf{R}^{-\frac{1}{2}}\mathbf{G}^{k-1}\mathbf{U} \end{pmatrix}. \tag{A6}$$

$\mathbf{f}(\delta\boldsymbol{v}^{k-1})$ and its Jacobian are fixed for each outer loop by the $k-1$ trajectory. Completing the GN algorithm, the Hessian

($\mathcal{H}_{\delta\boldsymbol{v}}$) is approximated by $\mathbf{A}' \equiv \mathbf{F}^{k-1\top}\mathbf{F}^{k-1}$, after ignoring mixed partial derivatives of $\mathbf{f}$. The Hessian of Eq. A2 matches

that of Eq. 10, namely

$$\mathcal{H}_{\delta\boldsymbol{v}} = \mathbf{I}_n + \mathbf{U}^\top\mathbf{G}^{k-1\top}\mathbf{R}^{-1}\mathbf{G}^{k-1}\mathbf{U}. \tag{A7}$$





After substitutions, Eq. A4 becomes

$$\mathbf{F}^{k-1^\top}\mathbf{F}^{k-1}\delta\boldsymbol{v}^k = -\mathbf{F}^{k-1^\top}\mathbf{f}\left(\delta\boldsymbol{v}^{k-1}\right), \tag{A8}$$

which expands to

$$\left(\mathbf{I}_n + \mathbf{U}^\top\mathbf{G}^{k-1^\top}\mathbf{R}^{-1}\mathbf{G}^{k-1}\mathbf{U}\right)\delta\boldsymbol{v}^k = \boldsymbol{d}^{b,k-1} + \mathbf{U}^\top\mathbf{G}^{k-1^\top}\mathbf{R}^{-1}\boldsymbol{d}^{o,k-1}. \tag{A9}$$

Solving for $\delta\boldsymbol{v}^k$ gives the same update formula that would result from setting Eq. 20 equal to zero,

$$\delta\boldsymbol{v}^k = \left(\mathbf{I}_n + \mathbf{U}^\top\mathbf{G}^{k-1^\top}\mathbf{R}^{-1}\mathbf{G}^{k-1}\mathbf{U}\right)^{-1}\left(\boldsymbol{d}^{b,k-1} + \mathbf{U}^\top\mathbf{G}^{k-1^\top}\mathbf{R}^{-1}\boldsymbol{d}^{o,k-1}\right). \tag{A10}$$

Thus, by defining $\mathbf{f}$ appropriately, the eqivalence between GN and incremental 4D-Var is verified.

**Appendix B: Derivation of the truncated inverse Hessian**

After $l$ inner iterations, the Lanczos vectors form an orthogonal matrix, $\mathbf{Q}_l = [\hat{\boldsymbol{q}}_1, .., \hat{\boldsymbol{q}}_l]$, which satisfies

$$\mathcal{H}_{\delta\boldsymbol{v}}\mathbf{Q}_l = \mathbf{Q}_l\mathbf{T}_l. \tag{B1}$$

The extremal eigenvalues of $\mathbf{T}_l$ are good approximations to $\mathcal{H}_{\delta\boldsymbol{v}}$'s extremal eigenvalues (Golub and Van Loan, 1996). $\mathbf{T}_l$ can be decomposed as

$$\mathbf{T}_l = \mathbf{W}_l\boldsymbol{\Lambda}_l\mathbf{W}_l^{-1}. \tag{B2}$$

If we were to carry out the minimization for $n$ steps, we would find all the Lanczos vectors, and would be able construct the full $\mathbf{T}$ and $\mathbf{Q}$ matrices. In that case, the orthogonal Lanczos vectors admit $\mathbf{Q}\mathbf{Q}^\top = \mathbf{I}$. When combined with Eq. B1,

$$\mathcal{H}_{\delta\boldsymbol{v}} = \mathbf{Q}\mathbf{W}\boldsymbol{\Lambda}\mathbf{W}^{-1}\mathbf{Q}^\top \tag{B3}$$

Because the eigenvectors are orthonormal,

$$\mathcal{H}_{\delta\boldsymbol{v}} = \left(\mathbf{Q}\mathbf{W}\right)\boldsymbol{\Lambda}\left(\mathbf{Q}\mathbf{W}\right)^\top. \tag{B4}$$

Thus, the eigenvectors of $\mathcal{H}_{\delta\boldsymbol{v}}$ are approximately equal to the normalized eigenvectors of $\mathbf{T}$, premultiplied by the matrix of Lanczos vectors, i.e.,

$$\mathcal{H}_{\delta\boldsymbol{v}} = \hat{\boldsymbol{\nu}}\boldsymbol{\Lambda}\hat{\boldsymbol{\nu}}^\top, \tag{B5}$$

where the $k_i{}^{th}$ eigenvector of $\mathcal{H}_{\delta\boldsymbol{v}}$ is

$$\hat{\boldsymbol{\nu}}_{k_i} = \mathbf{Q}\hat{\boldsymbol{w}}_{k_i}. \tag{B6}$$





The Hessian is constructed by

$$\mathcal{H}_{\delta v} = \hat{\boldsymbol{\nu}}\boldsymbol{\Lambda}\hat{\boldsymbol{\nu}}^{\top} = \sum_{k_i=1}^{n} \lambda_{k_i}\hat{\boldsymbol{\nu}}_{k_i}\hat{\boldsymbol{\nu}}_{k_i}^{\top}. \tag{B7}$$

Since the Hessian and its inverse have identical eigenvectors and reciprocal eigenvalues, the inverse is

$$[\mathcal{H}_{\delta v}]^{-1} = \sum_{k_i=1}^{n} \lambda_{k_i}^{-1}\hat{\boldsymbol{\nu}}_{k_i}\hat{\boldsymbol{\nu}}_{k_i}^{\top}. \tag{B8}$$

5 Although this expression is usable, computational resource limitations require $l << n$. Truncating the sum yields a low rank estimate for the inverse, and for the posterior error which it estimates.

   A more robust estimate of the posterior error is a low-rank update to the full-rank prior covariance, $\mathbf{B}$. To pursue that goal, first we return to the linear algebra formula, then add and subtract the identity matrix to get

$$\begin{aligned} \mathcal{H}_{\delta v} &= \mathbf{I} + \hat{\boldsymbol{\nu}}\boldsymbol{\Lambda}\hat{\boldsymbol{\nu}}^{\top} - \mathbf{I} \\ &= \mathbf{I} + \hat{\boldsymbol{\nu}}\boldsymbol{\Lambda}\hat{\boldsymbol{\nu}}^{\top} - \hat{\boldsymbol{\nu}}\mathbf{I}\hat{\boldsymbol{\nu}}^{\top} \\ &= \mathbf{I} + \sum_{k_i=1}^{n} (\lambda_{k_i} - 1)\hat{\boldsymbol{\nu}}_{k_i}\hat{\boldsymbol{\nu}}_{k_i}^{\top} \end{aligned} \tag{B9}$$

10 Now we repeat the truncation,

$$\mathcal{H}_{\delta v} \approx \mathbf{I} + \sum_{k_i=1}^{l} (\lambda_{k_i} - 1)\hat{\boldsymbol{\nu}}_{k_i}\hat{\boldsymbol{\nu}}_{k_i}^{\top}, \tag{B10}$$

where $\hat{\boldsymbol{\nu}}_{k_i}$ is constructed from the partial set of Lanczos vectors as

$$\hat{\boldsymbol{\nu}}_{k_i} = \mathbf{Q}_l\hat{\boldsymbol{w}}_{lk_i}. \tag{B11}$$

Next we apply the Sherman-Morrison formula to recursively build the inverse for each term in the sum. Throughout, we will
15 take advantage of the following two relationships for orthogonal vectors

$$\hat{\boldsymbol{\nu}}_j^{\top}\hat{\boldsymbol{\nu}}_j = 1$$

and

$$\hat{\boldsymbol{\nu}}_j^{\top}\hat{\boldsymbol{\nu}}_i = 0; \quad j \neq i.$$

   Starting with the first term,

$$\begin{aligned} \mathbf{N}_1^{-1} &= \left[\mathbf{I} + (\lambda_1 - 1)\hat{\boldsymbol{\nu}}_1\hat{\boldsymbol{\nu}}_1^{\top}\right]^{-1} \\ &= \mathbf{I}^{-1} - \frac{\mathbf{I}^{-1}(\lambda_1 - 1)\hat{\boldsymbol{\nu}}_1\hat{\boldsymbol{\nu}}_1^{\top}\mathbf{I}^{-1}}{1 + (\lambda_1 - 1)\hat{\boldsymbol{\nu}}_1^{\top}\mathbf{I}^{-1}\hat{\boldsymbol{\nu}}_1} \\ &= \mathbf{I} - \frac{(\lambda_1 - 1)\hat{\boldsymbol{\nu}}_1\hat{\boldsymbol{\nu}}_1^{\top}}{\lambda_1} \\ &= \mathbf{I} + \left(\lambda_1^{-1} - 1\right)\hat{\boldsymbol{\nu}}_1\hat{\boldsymbol{\nu}}_1^{\top}. \end{aligned}$$





This result fits our desired proof. Now for the second term,

$$
\begin{aligned}
\mathbf{N}_2^{-1} &= \left\{ \mathbf{N}_1 + (\lambda_2 - 1)\,\hat{\boldsymbol{\nu}}_2 \hat{\boldsymbol{\nu}}_2^\top \right\}^{-1} \\
&= \mathbf{N}_1^{-1} - \frac{\mathbf{N}_1^{-1}(\lambda_2 - 1)\,\hat{\boldsymbol{\nu}}_2 \hat{\boldsymbol{\nu}}_2^\top \mathbf{N}_1^{-1}}{1 + (\lambda_2 - 1)\,\hat{\boldsymbol{\nu}}_2^\top \mathbf{N}_1^{-1}\hat{\boldsymbol{\nu}}_2} \\
&= \mathbf{I} + \left(\lambda_1^{-1} - 1\right)\hat{\boldsymbol{\nu}}_1 \hat{\boldsymbol{\nu}}_1^\top - \\
&\quad \frac{\left[\mathbf{I} + \left(\lambda_1^{-1} - 1\right)\hat{\boldsymbol{\nu}}_1 \hat{\boldsymbol{\nu}}_1^\top\right](\lambda_2 - 1)\,\hat{\boldsymbol{\nu}}_2 \hat{\boldsymbol{\nu}}_2^\top \left[\mathbf{I} + \left(\lambda_1^{-1} - 1\right)\hat{\boldsymbol{\nu}}_1 \hat{\boldsymbol{\nu}}_1^\top\right]}{1 + (\lambda_2 - 1)\,\hat{\boldsymbol{\nu}}_2^\top \left[\mathbf{I} + \left(\lambda_1^{-1} - 1\right)\hat{\boldsymbol{\nu}}_1 \hat{\boldsymbol{\nu}}_1^\top\right]\hat{\boldsymbol{\nu}}_2} \\
&= \mathbf{I} + \left(\lambda_1^{-1} - 1\right)\hat{\boldsymbol{\nu}}_1 \hat{\boldsymbol{\nu}}_1^\top + \left(\lambda_2^{-1} - 1\right)\hat{\boldsymbol{\nu}}_2 \hat{\boldsymbol{\nu}}_2^\top .
\end{aligned}
$$

The dot products of orthogonal vectors cancels all terms in the numerator and denominator except the ones multiplied by the identity matrix. The same simplification applies to each additional sum, where the full sum can be expressed as

$$
\begin{aligned}
\mathbf{N}_l^{-1} = \mathbf{I} + \left(\lambda_1^{-1} - 1\right)\hat{\boldsymbol{\nu}}_1 \hat{\boldsymbol{\nu}}_1^\top - \\
\sum_{k_i=2}^{l} \frac{\left[\mathbf{I} + \sum_{r=1}^{k_i-1}\left(\lambda_r^{-1} - 1\right)\hat{\boldsymbol{\nu}}_r \hat{\boldsymbol{\nu}}_r^\top\right](\lambda_{k_i} - 1)\,\hat{\boldsymbol{\nu}}_{k_i} \hat{\boldsymbol{\nu}}_{k_i}^\top \left[\mathbf{I} + \sum_{r=1}^{k_i-1}\left(\lambda_r^{-1} - 1\right)\hat{\boldsymbol{\nu}}_r \hat{\boldsymbol{\nu}}_r^\top\right]}{1 + (\lambda_{k_i} - 1)\,\hat{\boldsymbol{\nu}}_{k_i}^\top \left[\mathbf{I} + \sum_{r=1}^{k_i-1}\left(\lambda_r^{-1} - 1\right)\hat{\boldsymbol{\nu}}_r \hat{\boldsymbol{\nu}}_r^\top\right]\hat{\boldsymbol{\nu}}_{k_i}}
\end{aligned}
$$

Here, again, all of the terms where $r \neq k_i$ cancel. What remains is similar to Eq. B8, but slightly modified.

$$
\left[\mathcal{H}_{\delta \boldsymbol{v}}\right]^{-1} \approx \mathbf{I} + \sum_{k_i=1}^{l} \left(\lambda_{k_i}^{-1} - 1\right)\hat{\boldsymbol{\nu}}_{k_i} \hat{\boldsymbol{\nu}}_{k_i}^\top . \tag{B12}
$$

After a left-side multiplication by $\mathbf{U}$ and a right-side multiplication by $\mathbf{U}^\top$, we achieve the desired low rank update to $\mathbf{B}$ found in Eq. 26.

*Author contributions.* J. J. Guerrette developed the software and ran simulations following the guidance of the PI, D. K. Henze. Both authors contributed to writing and analysis.

*Acknowledgements.* This research has been supported by a grant from the U.S. Environmental Protection Agency's Science to Achieve Results (STAR) program. Although the research described in the article has been funded wholly or in part by the U.S. Environmental Protection Agency's STAR program through grant R835037, it has not been subjected to any EPA review and therefore does not necessarily reflect the views of the Agency, and no official endorsement should be inferred. In addition, this paper is a result of research funded by the National Oceanic and Atmospheric Administration's Earth System Research Laboratory as part of the Fire Influence on Regional and Global Environments Experiment (FIREX) through the grant NOAA NA16OAR4310113. We are thankful for the ARCTAS mission, which was supported by NASA. We thank Y. Kondo for making the SP2 observations available through the NASA LaRC Airborne Science Data for Atmospheric Composition database. We acknowledge the use of FIRMS data from the Land Atmosphere Near-real time Capability for EOS (LANCE) system operated by the NASA/GSFC/Earth Science Data and Information System (ESDIS) with funding provided by NASA/HQ.



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

**Table 1.** Emission inversion scenarios.

| | Scenario | BB Inventory | $L_h$ | Obs. Used (day) |
|---|---|---|---|---|
| 22 June | FINN_STD | FINNv1.0 | 36 km | ARCTAS-CARB (22) |
| | FINN_L18 | FINNv1.0 | 18 km | ARCTAS-CARB (22) |
| | QFED_STD | QFEDv2.4r8 | 36 km | ARCTAS-CARB (22) |
| | QFED_L18 | QFEDv2.4r8 | 18 km | ARCTAS-CARB (22) |
| | FINN_V1.5 | FINNv1.5 | 36 km | ARCTAS-CARB (22) |
| 23/24 June | FINN_STD | FINNv1.0 | 36 km | IMPROVE (23) & ARCTAS-CARB (24) |
| | QFED_STD | QFEDv2.4r8 | 36 km | IMPROVE (23) & ARCTAS-CARB (24) |
| | ACFT | FINNv1.0 | 36 km | ARCTAS-CARB (24) |
| | SURF | FINNv1.0 | 36 km | IMPROVE (23) |





**Table 2.** Aircraft observation linear regression characteristics for the prior (background, b) and posterior (analysis, a).

| Obs. Date → | | 22 June, $N_{obs} = 241$ | | | | 24 June, $N_{obs} = 301$ | | | | 26 June, $N_{obs} = 117$ | | | |
|---|---|---|---|---|---|---|---|---|---|---|---|---|---|
| Inversion Scenario | | $R^2$ | | slope | | $R^2$ | | slope | | $R^2$ | | slope | |
| ↓ | | b | a | b | a | b | a | b | a | b | a | b | a |
| 22 June | FINN_STD | 0.11 | **0.82** | 0.26 | **0.80** | 0.18 | *0.15* | 0.38 | *0.25* | 0.56 | 0.52 | 0.15 | **0.49** |
| | QFED_STD | 0.03 | **0.73** | 0.34 | **0.71** | 0.15 | 0.23 | 0.43 | 0.37 | 0.59 | 0.53 | 0.39 | 0.43 |
| 23/24 June | FINN_STD | - | - | - | - | 0.17 | **0.52** | 0.35 | **0.56** | 0.59 | *0.16* | 0.15 | 0.11 |
| | QFED_STD | - | - | - | - | 0.11 | **0.52** | 0.36 | **0.55** | 0.63 | *0.44* | 0.41 | *0.15* |
| | ACFT | - | - | - | - | 0.17 | **0.53** | 0.35 | **0.57** | 0.59 | *0.29* | 0.15 | 0.08 |
| | SURF | - | - | - | - | 0.17 | 0.17 | 0.35 | 0.40 | 0.59 | *0.13* | 0.15 | 0.17 |

**distinct improvement**; *distinct degradation*; cross validation

**Table 3.** Surface observation linear regression characteristics for the prior (background, b) and posterior (analysis, a).

| Obs. Date → | | 23 June, $N_{obs} = 35$ | | | | 26 June, $N_{obs} = 36$ | | | |
|---|---|---|---|---|---|---|---|---|---|
| Inversion Scenario | | $R^2$ | | slope | | $R^2$ | | slope | |
| ↓ | | b | a | b | a | b | a | b | a |
| 22 June | FINN_STD | 0.06 | 0.04 | 0.26 | 0.21 | 0.03 | 0.05 | 0.10 | 0.13 |
| | QFED_STD | 0.16 | 0.14 | 0.44 | 0.41 | 0.10 | 0.11 | 0.20 | 0.21 |
| 23/24 June | FINN_STD | 0.04 | **0.75** | 0.25 | **1.04** | 0.03 | **0.28** | 0.10 | **0.28** |
| | QFED_STD | 0.09 | **0.74** | 0.39 | **1.01** | 0.09 | 0.15 | 0.20 | 0.16 |
| | ACFT | 0.04 | 0.05 | 0.25 | 0.27 | 0.03 | 0.03 | 0.10 | 0.09 |
| | SURF | 0.04 | **0.74** | 0.25 | **1.02** | 0.03 | **0.35** | 0.10 | **0.35** |

**distinct improvement**; *distinct degradation*; cross validation

**Table 4.** Emission area coordinates. EA1-4 are used for BB totals and EA5-9 are used for anthropogenic totals.

| | $LON_{min}$ | $LON_{max}$ | $LAT_{min}$ | $LAT_{max}$ |
|---|---|---|---|---|
| EA1 | 122.5°W | 120.5°W | 35.7°N | 38.5°N |
| EA2 | 123.8°W | 122.1°W | 38.9°N | 40.4°N |
| EA3 | 124.3°W | 122.9°W | 40.4°N | 41.7°N |
| EA4 | 122.1°W | 120.0°W | 38.5°N | 40.4°N |
| EA5 | 117.8°W | 116.9°W | 32.1°N | 33.4°N |
| EA6 | 121.0°W | 117.8°W | 33.4°N | 34.6°N |
| EA7 | 123.0°W | 121.0°W | 36.6°N | 38.8°N |
| EA8 | 120.6°W | 118.6°W | 35.2°N | 37.0°N |
| EA9 | 118.0°W | 116.5°W | 34.0°N | 36.0°N |
| EA10 | 116.9°W | 115.0°W | 32.1°N | 33.4°N |





**Table 5.** Total BB emissions for EA's and domain-wide during 22 and 23/24 June inversions (averaged for 24 hour period). Absolute units are in Mg. Note, the differences ($\Delta$) may not sum due to rounding.

| | | FINN_STD | | | QFED_STD | | | $\frac{\Sigma E_{QFED}}{\Sigma E_{FINN}}$ | |
| | | $\Sigma E_b$ | $\Sigma E_a$ | $\Delta$ | $\Sigma E_b$ | $\Sigma E_a$ | $\Delta$ | b | a |
|---|---|---|---|---|---|---|---|---|---|
| 22 June | EA1 | 14 | 4 | -10 | 82 | 26 | -55 | ×5.8 | ×6.4 |
| | EA2 | 6 | 30 | +24 | 9 | 15 | +6 | ×1.5 | ×0.5 |
| | EA3 | 6 | 4 | -2 | 29 | 7 | -22 | ×4.5 | ×1.6 |
| | EA4 | 18 | 22 | +4 | 52 | 83 | +31 | ×2.8 | ×3.8 |
| | DOMAIN | 59 | 83 | +34 | 209 | 171 | -38 | ×3.5 | ×2.1 |
| 23+24 June | EA1 | 20 | 5 | -15 | 70 | 12 | -58 | ×3.5 | ×2.5 |
| | EA2 | 28 | 11 | -16 | 96 | 29 | -67 | ×3.5 | ×2.6 |
| | EA3 | 17 | 12 | -5 | 37 | 20 | -17 | ×2.2 | ×1.7 |
| | EA4 | 32 | 108 | +77 | 107 | 107 | 0 | ×3.4 | ×1.0 |
| | DOMAIN | 138 | 249 | +111 | 471 | 354 | -117 | ×3.4 | ×1.4 |

**Table 6.** Total anthropogenic emissions for EA's and domain-wide during 22 and 23/24 June inversions (averaged for 24 hour period). The posterior for 23/24 June is from an inversion using both the IMPROVE and ARACTAS-CARB observations. Results shown are for the FINN_STD scenario. Absolute units are in Mg. Note, the differences ($\Delta$) may not sum due to rounding.

| | 22 June | | | 23/24 June | | | $\frac{\Sigma E_{23+24 June}}{\Sigma E_{22 June}}$ | |
| | $\Sigma E_b$ | $\Sigma E_a$ | $\Delta$ | $\Sigma E_b$ | $\Sigma E_a$ | $\Delta$ | b | a |
|---|---|---|---|---|---|---|---|---|
| EA5 | 5 | 5 | 0 | 7 | 3 | -3 | ×1.4 | ×0.7 |
| EA6 | 12 | 8 | -4 | 17 | 9 | -8 | ×1.4 | ×1.2 |
| EA7 | 10 | 6 | -5 | 16 | 8 | -8 | ×1.6 | ×1.5 |
| EA8 | 3 | 2 | -1 | 5 | 25 | +20 | ×1.6 | ×9.9 |
| EA9 | 5 | 4 | -1 | 6 | 11 | +4 | ×1.3 | ×2.7 |
| EA10 | 2 | 2 | 0 | 3 | 8 | +5 | ×1.4 | ×3.7 |
| DOMAIN | 81 | 68 | -13 | 114 | 123 | +9 | ×1.4 | ×1.8 |





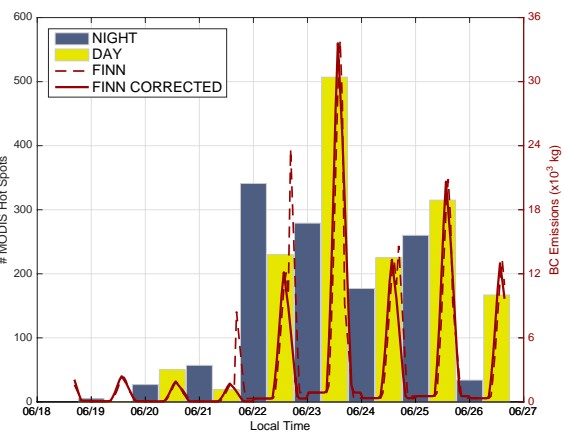

**Figure 1.** MODIS fire hot spot detections, excluding those with confidence less than or equal to 20% and double detections within 1.2 km of each other (left axis) and domain-wide FINNv1.0 BB emissions during the ARCTAS-CARB campaign, with and without fixes described in Sec. 2.2 (right axis).





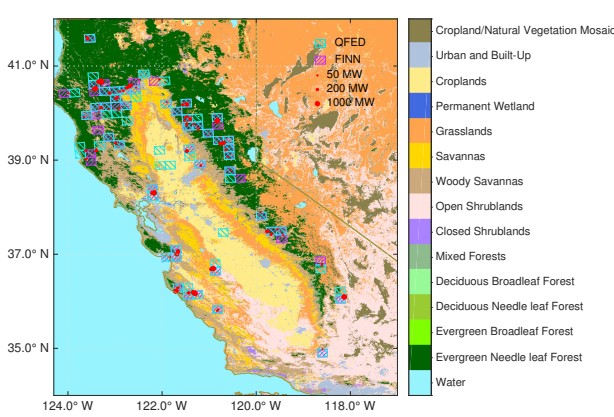

**Figure 2.** Land category types, MODIS fire hot spot detections on 21 and 22 June, 2008, sized by FRP, and 18 km×18 km gridded FINNv1.0 and QFED emission locations.





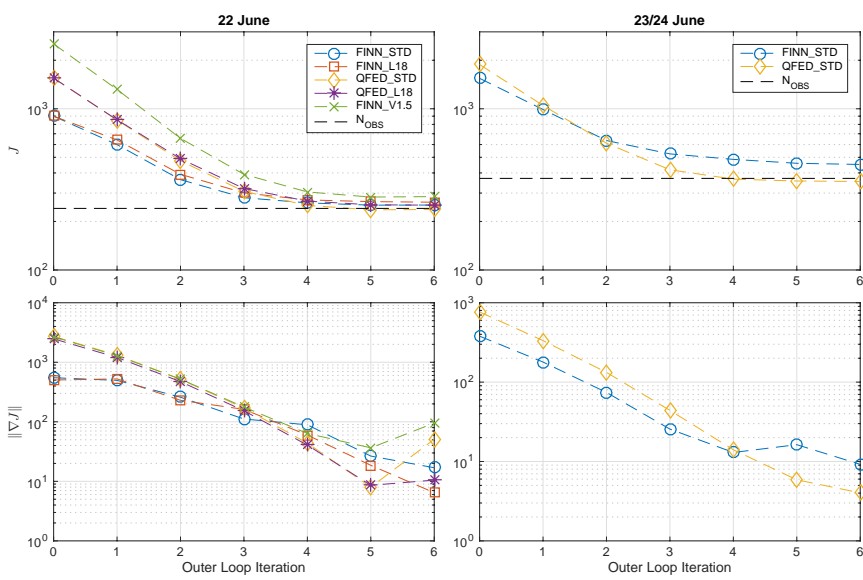

**Figure 3.** Outer loop cost function and gradient norm evaluations for the June 22 (left column) and 23/24 June (right column) inversions.





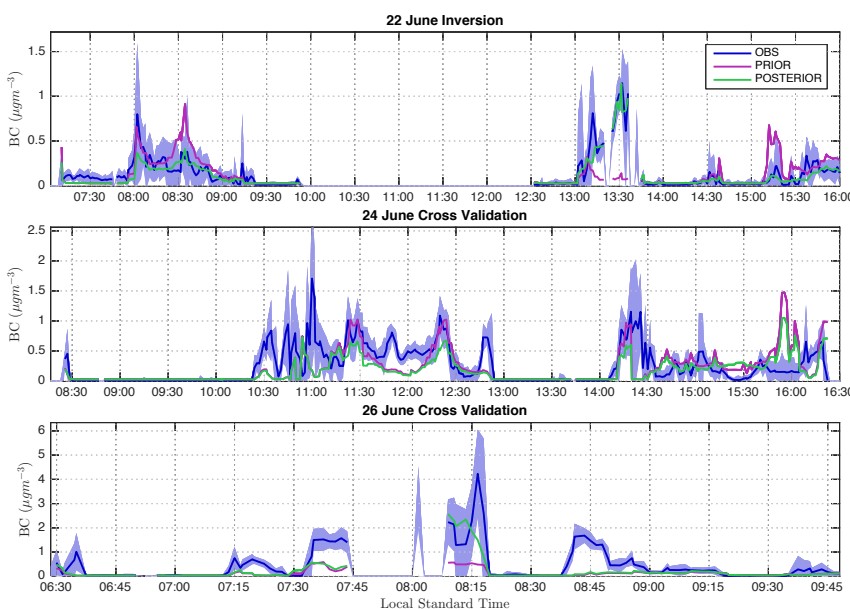

**Figure 4.** Temporal variation of observed, prior, and posterior BC concentrations during ARCTAS-CARB. The model values are obtained with the FINN_STD inversion scenario. The shaded area encompasses 2 standard deviations around the observations, which includes both model and observation uncertainty.





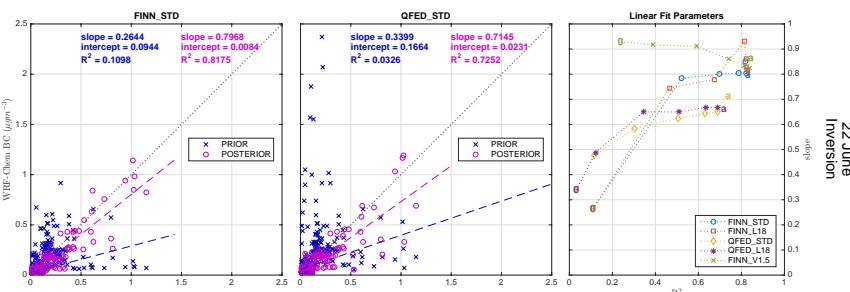

**Figure 5.** Prior and posterior model versus 22 June ARCTAS-CARB observations for the 22 June inversion. The right two columns are for FINN_STD and QFED_STD. Plots in the right column show the progression of slope and $R^2$ from the prior, "0", to the posterior, "a", for similar linear regressions in all scenarios.





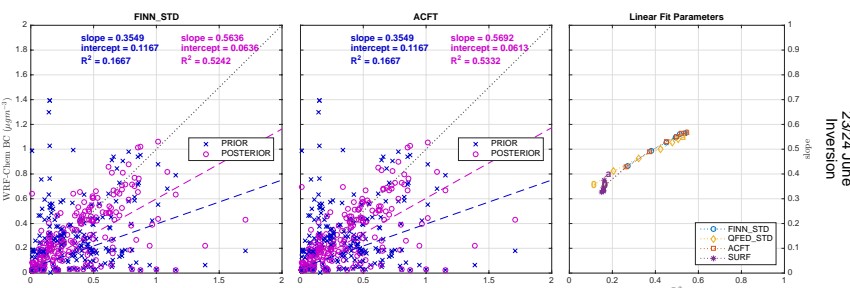

**Figure 6.** Prior and posterior model versus 24 June ARCTAS-CARB observations for the 23/24 June FINN_STD inversion. The left plot uses both IMPROVE (23 June) and ARCTAS-CARB observations in the inversion. The second column uses only ARCTAS-CARB. Plots in the right column show the progression of slope and $R^2$ from the prior, "0", to the posterior, "a", for similar linear regressions.





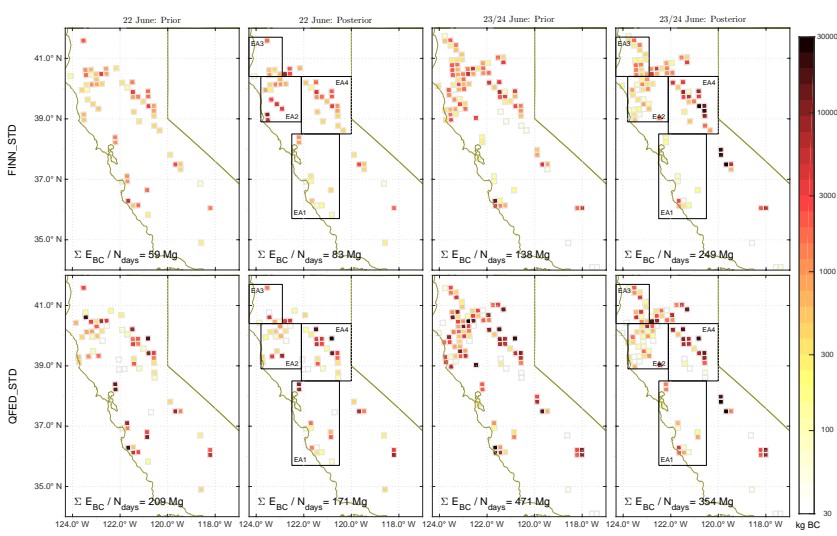

**Figure 7.** Prior and posterior grid-scale BB emissions of BC per 24 hours for FINN_STD and QFED_STD on 22 June, 00Z-23Z and 23 June, 00Z - 24 June 23Z. All emissions are expressed for a 24 h average. EA1-4 are outlined with black boxes.[NOTE on Figures 2, 7 and 8: we are waiting on results from QFED_STD with IMPROVE obs included.]





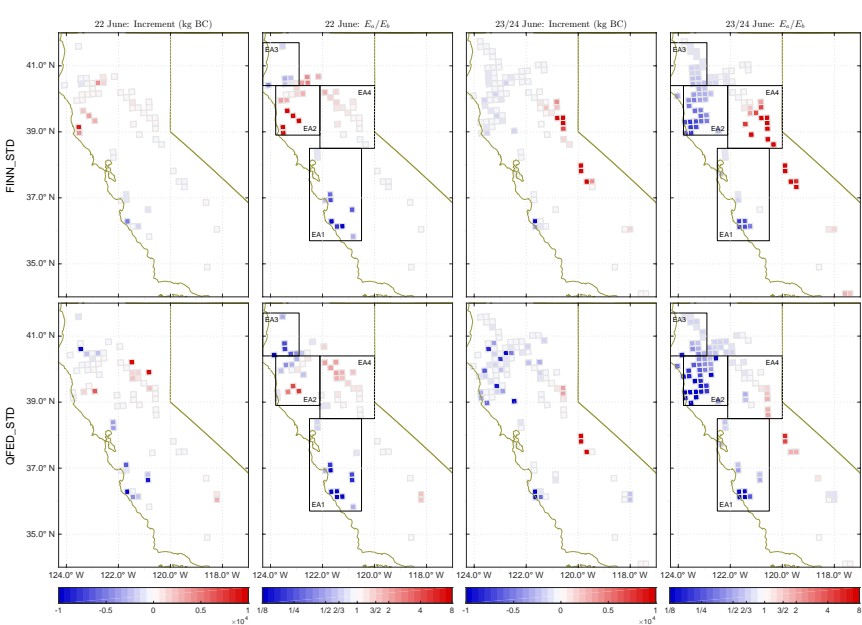

**Figure 8.** BB analysis increment (posterior minus prior) per 24 hours and posterior linear scaling factor ($\beta$) for the two primary BB scenarios on 22 June 00Z-23Z and 23 June, 00Z - 24 June 23Z. EA1-4 are outlined with black boxes. [NOTE on Figures 2, 7 and 8: we are waiting on results from QFED_STD with IMPROVE obs included.]





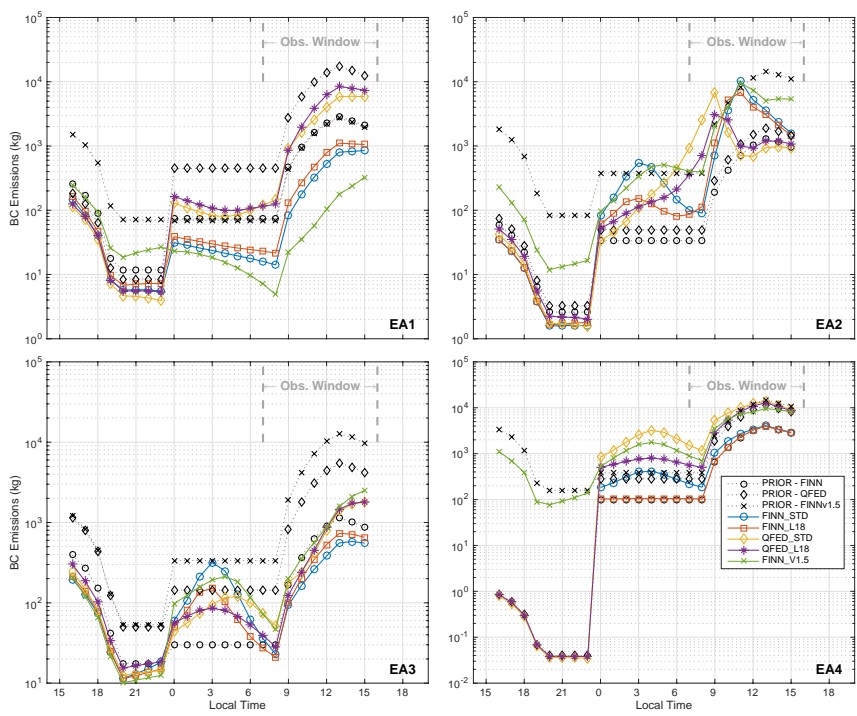

**Figure 9.** Hourly prior and posterior BB diurnal emission patterns for the four EAs and all inversion scenarios for 22 June, 00Z-23Z, with the time shown in LT. Note that FINNv1.0 did not have any fires in EA4 on 21 June.





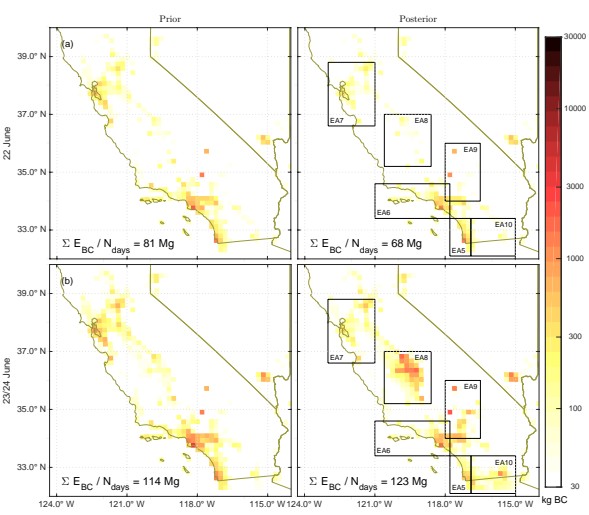

**Figure 10.** Prior and posterior grid-scale anthropogenic emissions of BC per 24 hours for FINN_STD on 22 June, 00Z-23Z (top row) and 23 June, 00Z to 24 June, 23Z. EA5-10 are outlined with black boxes.





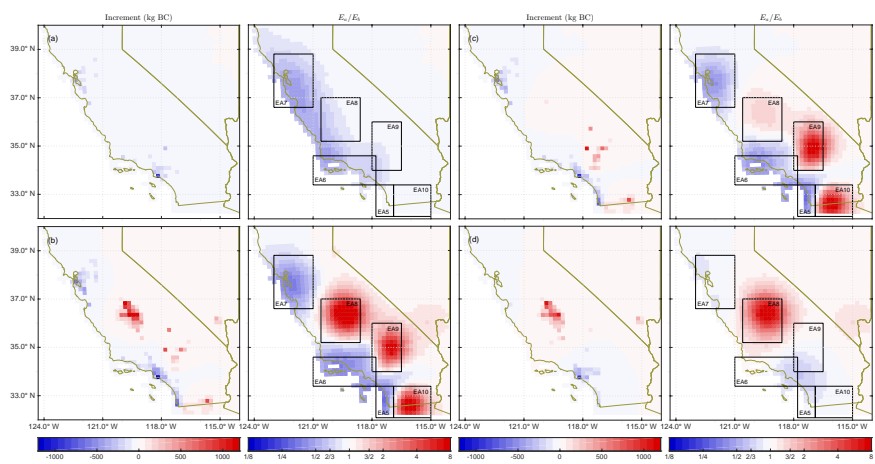

**Figure 11.** Anthropogenic analysis increment (posterior minus prior) per 24 hours and posterior linear scaling factor ($\beta$) for the (a) FINN_STD (22), (b) FINN_STD (23/24), (c) ACFT, and (d) SURF inversion scenarios. EA5-9 are outlined with black boxes in the scaling factor plots.





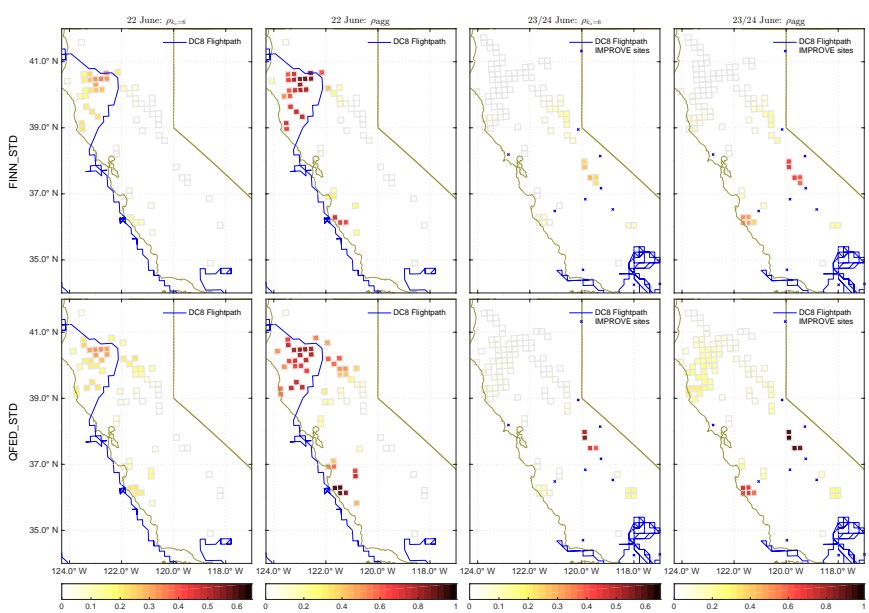

**Figure 12.** BB error reduction in the final outer loop ($\rho_{k_o=6}$) and aggregated across all outer loops ($\rho_{\mathrm{agg}}$) for the two primary BB scenarios on 22 June 00Z-23Z and 23 June, 00Z - 24 June 23Z. The ARCTAS-CARB DC8 flightpath and IMPROVE sites at model grid centers are overlaid.





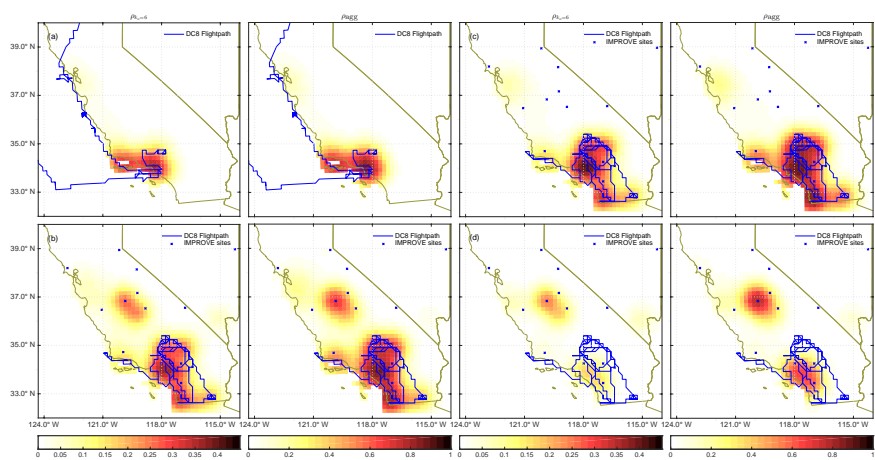

**Figure 13.** Anthropogenic error reduction in the final outer loop ($\rho_{k_o=6}$) and aggregated across all outer loops ($\rho_{\text{agg}}$) for the (a) FINN_STD (22), (b) FINN_STD (23/24), (c) ACFT, and (d) SURF inversion scenarios. The ARCTAS-CARB DC8 flightpath and IMPROVE sites at model grid centers are overlaid.





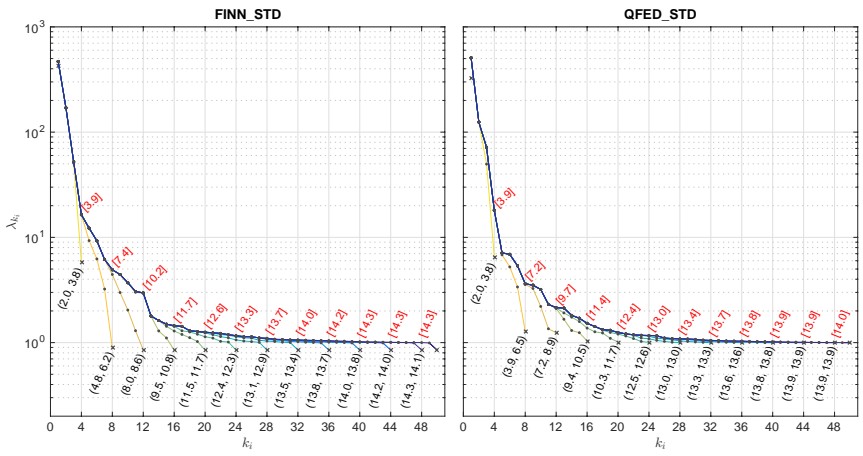

**Figure 14.** Eigenvalue spectra for FINN_STD and QFED_STD in the final outer loop on 22 June. The lines show the estimate of the spectrum $[\lambda_1, \dots, \lambda_{k_i=l}]$ in every fourth inner loop iteration, $l$. The black numbers in parentheses are the estimates of DOF that include eigenvalues in the sets (converged to within 5% of the previous estimate, all available). The red numbers in brackets are the truncated estimates of DOF using the most completely converged set of eigenvalues available in the 50[th] iteration.



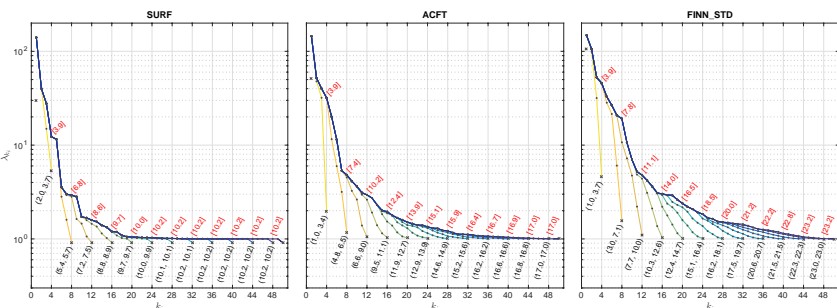

**Figure 15.** Eigenvalue spectra for SURF, ACFT, and SURF+ACFT in the final outer loop on 23 and 24 June. The lines show the estimate of the spectrum $[\lambda_1, \ldots, \lambda_{k_i=l}]$ in every fourth inner loop iteration, $l$. The black numbers in parentheses are the estimates of DOF that include eigenvalues in the sets (converged to within 5% of the previous estimate, all available). The red numbers in brackets are the truncated estimates of DOF using the most completely converged set of eigenvalues available in the 50th iteration.