# Peer review of "Four dimensional variational inversion of black carbon emissions during ARACTAS-CARB with WRFDA-Chem"

_Atmospheric Chemistry and Physics, 2016_

## Referee Comment (RC1) · Anonymous Referee #1 · 18 Nov 2016

This paper describes the use of a four dimensional data assimilation technique to assess the uncertainties in black carbon emissions over California. Extensive mathematical relationships associated with the adjoint technique are presented and they focus on two cases associated with the ARCTAS-CARB measurements to improve estimates of BC emissions. In general, the text is well written and the science sound; however, some of the motivation and conclusions are not articulated very well. In addition, I have a number of comments that need to be addressed before the manuscript is suitable for publication.

Major Comments: 1. Level of Detail: Extensive details associated with the mathematical relationships associated with four dimensional data assimilation will likely reduce

the overall target audience. While the air quality community is interested in methods to improve emission estimates, I would expect that much of those details will not be of much interest to the same community. More importantly, it was not clear whether the details on the adjoint are the same as those presented previously in the literature or whether they are new. If the relationships are typical of those of previous adjoint papers, perhaps more of the details could be put into the appendix. The authors should more clearly differentiate what is new and what is not new.

2. As described in a few of my specific comments, the discussion of what type of observations would be desirable to further constrain the adjoint technique and improve the emission estimates. The aircraft flights that targeted the fires were obviously critical, but would other flight paths be more useful? Or would more cases be useful? Were there only two periods during ARCTAS-CARB that were useful, or were the number of cases examined more limited by the computational expense of adjoint techniques?

3. The authors acknowledge that other meteorological factors will affect their analysis. They also mention that mostly clear skies were observed over California, so that complex cloud processes (i.e. wet removal) did not occur in this study. Near the end of the paper they mention it would be useful to have even simpler meteorological conditions to reduce uncertainties in meteorology. But it would seem very difficult to find such cases and one would have to confront complex real-world conditions at some point anyway.

Specific Comments:

Title: ARCTAS is misspelled.

Page 1, line 7: Consider changing "multiple" to "three". The next line lists 3 inventories.

Page 1, line 9: Change the use of "x" and through out the text to write out what they actually mean in terms of a change. I find the usage in this particular sentence to be confusing.

Page 3, lines 25-35: The motivation of why ARTCAS-CARB campaign is used for their

analysis should be improved. The way the paragraph is phrased, it basically just says they are going to use this particular campaign. But this could be changed to upfront state that the campaign had aircraft measurements characterizing both anthropogenic and biomass burning sources of BC and therefore would be useful to test their adjoint-based technique.

Page 4, line 17: Change "also turned off" to something else. Since it is not available, it is not possible to turn off that option. Also large fires could significantly affect meteorology by dramatically reducing incoming shortwave radiation, so some of the uncertainties in the adjoint technique will be due to this process that is neglected – in addition to the other meteorological processes they note.

Page 4,line 32: The grid spacing of 18 km is rather coarse, especially in resolving terrain-induced circulations in California. There have been numerous studies on this subject for California, and it would be useful to point that out. Although not explicitly stated here, the choice of coarse grid spacing is likely due to the computation cost of the adjoint-based technique.

Page 5, line 5: I could be wrong, but I thought the FINN emissions provided emissions per fire (i.e. point) that did not provide a spatial information on the size of the fire.

Page 5, line 13: I am skeptical of scaling the AOD based on a known high bias in GEOS-5. The bias is not necessarily linearly related to emissions. The problem in AOD in GEOS-5 could be a host of issues, such as representing the right mix of scattering and absorbing aerosols, water uptake of aerosols, and the treatment of aerosol optical properties.

Page 6, lines 5 – 9: Figure 2 shows the location of the MODIS fires, but the discussion is more about the shifts in the fires in the datasets. The text somehow implies Figure 2 illustrates those shifts, which it does not. Please change the text to clarify.

Page 14, line 9:the use of "swept out" should be changed. Implies the winds pushed

the aircraft over the ocean as opposed to a choice by the scientists or aircraft crew.

Page 14, lines 3 – 14: Suggest including the aircraft flight paths in Figure 2 to illustrate the discussion in this paragraph.

Page 14, line 18: Do you mean "Average" or "re-average"? They mention the 10-s data in the previous sentence, so perhaps that is a 10-s average? If that is the case, the text does not say so. The 90-s averaging is acknowledging the mismatch between the observations and the temporal resolution of the model. What I found missing was a discussion on the mismatch in spatial resolution, or does that even matter in the adjoint technique used? The model grid cell is 18 km, so it cannot resolve small-scale variation, so shouldn't the data be averaged to 18 km intervals?

Page 15, line 18: The authors mention that their adjoint technique requires 600 more computational time than a single simulation. It is good that this is mentioned. What is not included how many man-months or man-years such an effort requires (and that does not need to be included). But I would like to see some discussion at the end of the paper to weigh in on the advantages and disadvantages of the additional computation cost. In other words, is it worth the effort?

Page 15, lines 22 – 26: In this section of the details of the adjoint technique, I found it difficult to understand why it is important to show the convergence properties. I found the answer a few lines down on line 26. This rationale is a bit buried. I found the other sections had similar problems in terms of why it was important to visit aspects of the adjoint technique. This is a reflection that the authors assume the audience has a detailed knowledge of data assimilation and the adjoint tecnhnique in particular. So I think the concepts could be better communicated to a larger audience.

Page 16, line 12: In relation to "overprediction seems to be less of a problem" to me is a result of the coarse grid spacing. Using dx =18 km, the model should underpredict the peak concentrations of the emissions are correct. Only when the grid spacing is finer will the model resolve details of the smoke plume and there will be periods in which the

concentrations could be higher than observed.

Page 16, line 35: A 3.8 factor of uncertainty is used for the emissions. Did the authors try using values larger then 3.8?

Page 18, lines 3 – 11: The increases in BC emissions in the Fresno area are remarkable, so that it seems to have higher emissions than the LA area (at least from the scale in the figure). The explanation for missing BC emission seems plausible, but would missing railroad emission produce values as high as the entire LA basin? Does this seem realistic?

Page 21, lines 17-18: Isn't the first phrase in this sentence an obvious one that does not require this study to point out?

Page 22, lines 9 – 15: I am not sure what to make with the conclusion in the last sentence. It is true that in the cases examined in this study the aircraft flight paths and surface data might be saying different things about sources (since they are likely decoupled from one another). On other days that might not be the case when the aircraft flies in the boundary layer and will be more similar toe h surface measurements. The concluding sentence seems to contract the whole notion of using ARCTAS-CARB for their test of the adjoint technique. However, a large fraction of fires, as well as field campaign data to characterize fires are conducted over forest regions that are often located in areas of complex terrain. It seems to be a problem one has to confront.

Page 22: line 27: The authors state upfront and here that the paper focused on emissions only and not other factors that can affect BC concentrations. In the future, will they extend the analysis to meteorology to reduce those factors as well?

Conclusion: There are a few points I find missing or poorly articulated. The first is related to recommendation on future sampling. Here and there in the paper the authors mention or allude to changes in the sapling strategy that would help in better constraining the emission. It would be useful to include a summary of X ,Y, and Z they

feel would be useful. Second, how many more cases would be needed to have more robust estimates of the emissions? Finally, another discussion I mentioned in an early comment, is the computational cost and effort of using an adjoint technique worth the cost? One could take an alternative approach is to simply perform a small number of sensitivity simulation that scale the emissions to get a better fit. This is of contrast to a brute force method, which is not as mathematically pleasing as the adjoint.

Figure 2: The colors used for the vegetation distribution make it difficult to see the red dots denoting the fires. It would be useful to include the aircraft flight paths.

Figure 7: Consider changing the color scale, it is very difficult to see changes from one figure to the next.

Figure 9: The caption denotes posterior BB, but it is not clear which lines in the figure are posterior based on the legend.

---

## Referee Comment (RC2) · Anonymous Referee #3 · 11 Feb 2017

The authors estimated biomass burning and anthropogenic black carbon aerosol emissions based on proven 4D-Var technique that used in the WRFDA. The authors developed the forward, adjoint, and tangent linear models to calculate innovation, gradient, and cost function. The work presented in this study is very soundful. Also, Developing adjoint model is one of the most challenge work in aerosol data assimilation especially for assimilate satellite data. The presented results shows that the method would improve the chemical inversion performance via cross validation. However, the authors need some extra analyses/explanations before publication.

(1)One major concern of the MS is that the numerical experiments were only based on two real case forecasting. Since the atmospheric chemistry and meteorological

conditions vary day to day. It is suggested that the authors extend the experiments.

(2)How many numbers of the control variables in your CHEMDA system? Only BC? Please clarify the observation Operator and its adjoint. If the control variable 'same' as observation, the observation Operator and adjoint is simply as interpolation method. If not, you should clarify in the manuscript. In addition, how to deal with cross correlation between control variables (e.g. NO3 is correlate with SO4) if the number of CV is more than two.

(3)Incremental method is commonly used method in data assimilation. The manuscript appear 'incremental' many times. I suggest delete redundant 'incremental' in the manuscript.

(4)Background error covariance (BEC) is important in data assimilation. The observation information spread to model grid cells via BEC. The authors mentioned chemical emissions heterogeneous. However, the construction of the 'B' in the manuscript seems to be homogeneous?

---

## Author Comment (AC1) · 24 Mar 2017

We thank the referees for their reviews, which were instrumental in making a much improved manuscript. In response, we have made changes where we felt they were warranted, and gave replies to all comments. In this response, all of the page and line numbers for corrections refer to the revised manuscript, while we keep page and line numbers intact from the original reviews. The manuscript diff document is consistent with the changes described below.

**1 Responses to Anonymous Referee #1**

1. *MAJOR COMMENT:Level of Detail: Extensive details associated with the mathematical relationships associated with four dimensional data assimilation will likely reduce the overall target audience. While the air quality community is interested in methods to improve emission estimates, I would expect that much of those details will not be of much interest to the same community. More importantly, it was not clear whether the details on the adjoint are the same as those presented previously in the literature or whether they are new. If the relationships are typical of those of previous adjoint papers, perhaps more of the details could be put into the appendix. The authors should more clearly differentiate what is new and what is not new.*

   **AUTHOR RESPONSE:** We have reorganized the Introduction and Section 2.3 extensively and added clarifying introductions and transitions to guide the general audience. Although we still retain significant mathematical details, these are limited to those necessary for understanding how this approach is different than previous ones, how the results are obtained, and what those results mean. The unique challenge of the theoretical portion of this work was in applying an additive preconditioned incremental 4D-Var approach to log-normal control variables. Mutliplicative incremental 4D-Var for log-normal control variables was discussed in detail by Fletcher and Jones (2014).

   **Manuscript changes:**

   The Introduction has been modified to include the following: "The modifications to that system that are required for this work are described in Sec. 2 as well as in Guerrette and Henze (2015) (GH15). These include new linearized model descriptions (GH15), memory and I/O trajectory management (GH15), a log-normal emission control variable (Sec. 2.3.2), calculation of posterior variance (Sec. 2.3.4), and improvements to the Gauss-Newton optimization algorithm to handle nonlinearities (Sec. 2.3.5). As described in GH15, this approach of assimilating chemical tracer observations in a regional numerical weather prediction and chemistry model is unique in the context of previous 4D-Var flux constraints."

   We combined the opening to Section 2.3 with Sections 2.3.1 and 2.3.3 in a more logical flow. We now start new Section 2.3.1 with incremental 4D-Var instead of the more fundamental derivation that is well-known from previous literature. p12, line 2: added a sentence for clarity on posterior covariance; "While areas where uncertainty has been reduced from the prior include new information from the observations, areas without uncertainty reduction are simply a new realization of the prior."

Section 2.3.2 is split. One half is merged with the description of lognormal emissions (new Sec. 2.3.2). The other is separated into its own subsection that deals with Gaussian errors (new Sec. 2.3.3). The subsections are also reorganized and include new transition sentences to help clarify why we are discussing specific details and also what is new versus old. Much of the details in section 2.3.5 have now been removed.

2. *MAJOR COMMENT:As described in a few of my specific comments, the discussion of what type of observations would be desirable to further constrain the adjoint technique and improve the emission estimates. The aircraft flights that targeted the fires were obviously critical, but would other flight paths be more useful? Or would more cases be useful? Were there only two periods during ARCTAS-CARB that were useful, or were the number of cases examined more limited by the computational expense of adjoint techniques?*

**AUTHOR RESPONSE:**

There were four flights during the California portion of ARCTAS (20, 22, 24, and 26 June). The 20 June flight characterized Northern California anthropogenic sources before the fires started. The 26 June flight only flew over California for 3 hours, and then transited to the next base of operations. The 22 and 24 June observations were across longer durations and influenced significantly by BB sources. There was no underlying limitation of 4D-Var that prevented using the other flights, but those observations were limited in terms of the amount of information they contained about BB sources, which was our primary interest. We added some of these details to the manuscript.

**Manuscript changes:**

pp13, line 21: Addition; "The 20 June flight of ARCTAS-CARB characterized Northern California anthropogenic sources, but was not influenced by fires."

pp13, line 24: Modification; "The 24 June flight passed back and forth in the downwind region between Los Angeles and San Diego, measuring the outflow from those cities and the transportation between them, and 1 day old diluted BB outflow from the north."

3. *MAJOR COMMENT:The authors acknowledge that other meteorological factors will affect their analysis. They also mention that mostly clear skies were observed over California, so that complex cloud processes (i.e. wet removal) did not occur in this study. Near the end of the paper they mention it would be useful to have even simpler meteorological conditions to reduce uncertainties in meteorology. But it would seem very difficult to find such cases and one would have to confront complex real-world conditions at some point anyway.*

**AUTHOR RESPONSE:** Please see response to "Specific Comment" 19

1. *Specific Comment: Title: ARCTAS is misspelled.*

**Manuscript changes:** The title is corrected.

2. *Specific Comment: Page 1, line 7: Consider changing "multiple" to "three". The next line lists 3 inventories.*

**Manuscript changes:** We changed "multiple" to "three".

3. *Specific Comment: Page 1, line 9: Change the use of "×" and through out the text to write out what they actually mean in terms of a change. I find the usage in this particular sentence to be confusing.*

   **AUTHOR RESPONSE:** We agree that several of the instances of "×" are difficult to comprehend. We added terminology in both the abstract and the first paragraph to introduce × to mean "a factor of", as in a factor of 2 to 3 (×2 to ×3. Anywhere this exact phrasing does not fit, we have fixed the text. We also rearranged some other sentences with this vernacular for ease of reading.

   **Manuscript changes:** The following sentences are modified:

   ORIGINAL p1,line9: p1, line 9: "On 22 June, aircraft observations are able to reduce the spread between a customized QFED inventory and FINNv1.0 from a factor of 3.5 (×3.5) to only ×2.1."

   p2, line 10: "Zhang et al. (2014a) concluded that diffusion and loss mechanisms limit the corresponding responses of domain-wide aerosol burden, AOD, and 2 m temperature to ×2-3."

   p2, line 4: "Bond et al. (2013) cite several inventories of annual U.S. non-BB BC sources, which are between 260 to 440 Gg yr$^{-1}$, yielding a maximum to minimum ratio of 1.7."

   p14, line 33: "...plus the adjoint (×10 longer than the nonlinear model), the cost of incremental 4D-Var is approximately ×600 more than that of a single forward simulation..."

4. *Specific Comment: Page 3, lines 25-35: The motivation of why ARTCAS-CARB campaign is used for their analysis should be improved. The way the paragraph is phrased, it basically just says they are going to use this particular campaign. But this could be changed to upfront state that the campaign had aircraft measurements characterizing both anthropogenic and biomass burning sources of BC and therefore would be useful to test their adjoint-based technique.*

   **Manuscript changes:** We divided this long paragraph into three smaller ones, and add language to clarify why we used ARCTAS-CARB.

5. *Specific Comment: Page 4, line 17: Change "also turned off" to something else. Since it is not available, it is not possible to turn off that option. Also large fires could significantly affect meteorology by dramatically reducing incoming shortwave radiation, so some of the uncertainties in the adjoint technique will be due to this process that is neglected — in addition to the other meteorological processes they note.*

   **Manuscript changes:** p4, line 3: Modification; "Microphysical and radiative responses to online aerosols are not taken into account for GOCART aerosols in WRF-Chem."

6. *Specific Comment: Page 4,line 32: The grid spacing of 18 km is rather coarse, especially in resolving terrain-induced circulations in California. There have been numerous*

*studies on this subject for California, and it would be useful to point that out. Although*
*not explicitly stated here, the choice of coarse grid spacing is likely due to the compu-*
*tation cost of the adjoint-based technique.*

**AUTHOR RESPONSE:** We added the following paragraph as this is an impor-
tant consideration for future work. Please see the updated document for appropriate
references.

**Manuscript changes:** "Our horizontal grid spacing was chosen to balance the wall-
time and memory requirements of 4D-Var with model accuracy, and the ACM2 PBL
option was chosen to reduce ADM and TLM development efforts. Angevine et al.
(2012) recommend that the complex terrain in California demands fine tuning of the
WRF horizontal grid spacing, PBL, LSM, and reanalysis initialization. Among other
conclusions, those authors found that at six surface sites near the land-ocean boundary
a 4 km and a 12 km simulation with similar settings had mean wind speed biases of (0.15
to 1.5) $\mathrm{m\,s^{-1}}$ and (-0.38 to 1.9) $\mathrm{m\,s^{-1}}$, respectively. Supporting that conclusion, Strand
et al. (2012) used a 36 km resolution chemical transport model (CTM), with offline
meteorology, and found significant negative mean fractional bias (MFB) in modeled
$PM_{2.5}$ relative to surface observations of fires within narrow Northern California valleys
in July 2008 (MFB=-34.95%) and during autumn 2007 Santa Ana winds (MFB=-
110.22%). During the July 2008 episode, their CTM predictions had a smaller positive
bias (MFB=+21.88%). Therefore, we would expect similar wind and concentration
biases at 18 km resolution, which may or may not be improved by online meteorology.
Incremental 4D-Var provides an opportunity to utilize a different model configuration
(e.g., resolution) for the NLM comparisons of model to observations than that used for
the ADM and TLM simulations. The adaptation of that capability from meteorological
(i.e., Zhang et al., 2014b) to chemical simulations and the subsequent testing is reserved
for future WRFDA-Chem developments."

7. *Specific Comment: Page 5, line 5: I could be wrong, but I thought the FINN emissions*
*provided emissions per fire (i.e. point) that did not provide a spatial information on*
*the size of the fire.*

**AUTHOR RESPONSE:** According to Wiedinmyer et al. (2011), each fire source is
assumed to cover the entire 1 km × 1 km MODIS pixel from which the fire detection
was captured. This area (1 km$^2$) is assumed equal to the burned area for that day,
which is used to calculate emissions. There is an exception for grassland/savannas,
which has an assumed area of 0.75 km$^2$, and also when the MODIS VCF product
includes non-zero fractional bare cover for a pixel.

8. *Specific Comment: Page 5, line 13: I am skeptical of scaling the AOD based on a known*
*high bias in GEOS-5. The bias is not necessarily linearly related to emissions. The*
*problem in AOD in GEOS-5 could be a host of issues, such as representing the right*
*mix of scattering and absorbing aerosols, water uptake of aerosols, and the treatment*
*of aerosol optical properties.*

**AUTHOR RESPONSE:** We also don't believe that the GEOS-5 AOD scaling should
give perfect emissions. Still, we wanted to compare these inventories following the methodologies proposed by their creators to see if one has an advantage over the
other. Although we made several modifications when inconsistencies arose with either
observations (AOD for QFED) or known environmental behavior (diurnal pattern of
sunlight), we feel that those fixes were not outside the intentions of those developers.
We added the following paragraph to clarify the corrections we made to the inventories.

**Manuscript changes:** p5, line 29-32: Addition; "Any inverse modeling study that
depends on the first guess should start in a region of high probability. In a Bayesian
inversion, the first guess should be unbiased on average. Here we address several
known errors in our prior inventories that we either fix or are unable to fix. All
changes are consistent with either observations or the intended physical descriptions
of the inventories."

9. *Specific Comment: Page 6, lines 5-9: Figure 2 shows the location of the MODIS fires,*
*but the discussion is more about the shifts in the fires in the datasets. The text somehow*
*implies Figure 2 illustrates those shifts, which it does not. Please change the text to*
*clarify.*

**AUTHOR RESPONSE:** We chose poor colors for Figure 2. In the new figure, the
difference between FINN and QFED gridded emissions is much clearer, especially due
to the zoomed in map.

**Manuscript changes:** Replaced Figure 2.

10. *Specific Comment: Page 14, line 9:the use of "swept out" should be changed. Implies*
*the winds pushed the aircraft over the ocean as opposed to a choice by the scientists or*
*aircraft crew.*

**Manuscript changes:** p13, line 21: Modification; " disembarked from Los Angeles,
swept out over the ocean" changed to "embarked from Los Angeles, transited the
off-shore Pacific inflow"

11. *Specific Comment: Page 14, lines 3-14: Suggest including the aircraft flight paths in*
*Figure 2 to illustrate the discussion in this paragraph.*

**AUTHOR RESPONSE:** The flightpath is added.

**Manuscript changes:** Replaced Figure 2.

12. *Specific Comment: Page 14, line 18: Do you mean "Average" or "re-average"? They*
*mention the 10-s data in the previous sentence, so perhaps that is a 10-s average? If*
*that is the case, the text does not say so. The 90-s averaging is acknowledging the*
*mismatch between the observations and the temporal resolution of the model. What*
*I found missing was a discussion on the mismatch in spatial resolution, or does that*
*even matter in the adjoint technique used? The model grid cell is 18 km, so it cannot*
*resolve small-scale variation, so shouldn?t the data be averaged to 18 km intervals?*

**AUTHOR RESPONSE:** We intended to say "re-average". That procedure is de-
scribed in Section 5.1.2 of Guerrette and Henze (2015): "The 10 s resolution ARCTAS
observations of BC concentration, pressure, latitude, and longitude are averaged to
the 90 s model time step, which is approximately the time the DC-8 would take to traverse a single 18 km × 18 km column. However, the 10 s resolution ARCTAS BC concentrations are revision 2 (R2), while a later revision 3 (R3) product was released at 60 s resolution only. The later revision includes additional mass in the 50-900 nm size range as a result of applying a lognormal fit. In order to utilize this improved product, as well as leverage the finer resolution observations, the 10s BC mass is scaled by the mass ratio between the 60 s R3 and the 60 s average R2 data sets. The scaled 90 s average observations are compared directly with the nearest model grid cell so that the model values are not interpolated."

The temporal (not spatial) averaging and nearest-neighbor approaches introduce some mismatch between the observation and adjoint forcing locations. Spatial interpolation would reduce that error somewhat. We have not tested the impact on the posterior, but at least the adjoint spatial dispersion should enforce some correction.

13. *Specific Comment: Page 15, line 18: The authors mention that their adjoint technique requires 600 more computational time than a single simulation. It is good that this is mentioned. What is not included how many man-months or man-years such an effort requires (and that does not need to be included). But I would like to see some discussion at the end of the paper to weigh in on the advantages and disadvantages of the additional computation cost. In other words, is it worth the effort?*

**AUTHOR RESPONSE:** An objective answer to this question would require an extensive meta analysis of the benefits and costs of adjoint-based 4D-Var compared to other approaches, and would need to examine the valuation of labor versus hardware . But hopefully the results of this work could contribute to such a study someday. Also, for clarification we note that a single adjoint simulation in WRFPLUS-Chem requires a wall-time that is approximately 7 times that of a single forward model simulation, not 600 times. It is the optimization (4D-Var) that requires most of the extra resources.

14. *Specific Comment: Page 15, lines 22-26: In this section of the details of the adjoint technique, I found it difficult to understand why it is important to show the convergence properties. I found the answer a few lines down on line 26. This rationale is a bit buried. I found the other sections had similar problems in terms of why it was important to visit aspects of the adjoint technique. This is a reflection that the authors assume the audience has a detailed knowledge of data assimilation and the adjoint tecnhnique in particular. So I think the concepts could be better communicated to a larger audience.*

**AUTHOR RESPONSE:** This paragraph and the following one have been reorganized for clarity and accuracy. In particular, the topic sentence for the following paragraph was previously misleading for the text within. There was also a sentence about model uncertainty being higher in locations with higher prior concentrations. While that is true on average, it is not the dominant error behavior corresponding to 8:00 LT and 8:30 LT on 22 June, where the difference in observation uncertainty is more evident due to the difference in observed concentration.

**Manuscript changes:** See p15, lines 2-23.

15. *Specific Comment: Page 16, line 12: In relation to "overprediction seems to be less of a problem" to me is a result of the coarse grid spacing. Using dx =18 km, the model*

*should underpredict the peak concentrations of the emissions are correct. Only when the grid spacing is finer will the model resolve details of the smoke plume and there will be periods in which the concentrations could be higher than observed.*

**AUTHOR RESPONSE:** While originally we thought that the observation smoothing should be enough to counteract this resolution problem, we now realize a mistake in that logic. We modify the text as described below.

**Manuscript changes:** We removed that sentence and added the following paragraph (p16, lines 1-9): "For the appreciable measured BC concentrations ($> 0.25\,\mu g\,m^{-3}$), that are likely caused by a source within the model domain and simulation period, the lack of a source-receptor relationship is likely caused by low resolution. Changing a point source to a grid-scale area source changes its effective location. Temporal averaging of the observations will not necessarily solve that problem since perfectly modeled transport could still send a mis-located source in an entirely different direction than the truthfully located source. This effect is evident for valley fires (Strand et al., 2012), since placing the sources in the basin or spreading them throughout the basin and the peaks will result in different "downwind" concentrations. Downwind might be a very different direction if the convective scale winds contribute more information than the mesoscale winds to the true source-receptor relationship. Since the emissions are smoothed in the model and not in reality, the mis-location is more likely to cause under-prediction than over-prediction."

The coarse resolution offsets the location of the sources relative to the fire detections (Figure 2), and this would prevent the true sparse BB point sources from aligning with the model-observation residual error. However, the observed intensity of the true smoke plume is muted by the temporal averaging. The posterior model concentrations in Fig. 4 (top) mimic the temporal resolution of those observations. Thus, the model could achieve the same intensity as the observations with correct emission magnitudes and locations.

16. *Specific Comment: Page 16, line 35: A 3.8 factor of uncertainty is used for the emissions. Did the authors try using values larger then 3.8?*

**AUTHOR RESPONSE:** That value was not arbitrary (see p14, lines 17-23). We tried values up to a factor of 10 for biomass burning sources in FINN_STD and QFD_STD, but larger uncertainties tended to raise the value of the final cost function. Since FINN V1.5 is an outlier in this location and at several others, it warrants a larger uncertainty.

17. *Specific Comment: Page 18, lines 3-11: The increases in BC emissions in the Fresno area are remarkable, so that it seems to have higher emissions than the LA area (at least from the scale in the figure). The explanation for missing BC emission seems plausible, but would missing railroad emission produce values as high as the entire LA basin? Does this seem realistic?*

**AUTHOR RESPONSE:** To make this statement less definative, we have made the following modifications.

**Manuscript changes:** (1) p17, line 35 to p18, line 2: Modification; "it could be speculated that the prior is missing diesel rail sources of BC. Another possibility..." changed to "the inversion results may suggest that the prior is missing diesel rail sources of BC. However, for locations where the prior magnitude of BB and anthropogenic emissions are of similar magnitude, their posteriors are subject to projection from one sector to another. It is more likely ..."

18. *Specific Comment: Page 21, lines 17-18: Isn't the first phrase in this sentence an obvious one that does not require this study to point out?*

**AUTHOR RESPONSE:** What we found important to point out is that alternative observing strategies are required to characterize emissions. This is not surprising since IMPROVE was designed specifically to characterize background concentrations. Showing that this tool can at least tease out that basic observing strategy is the first step toward using such methods to plan more elaborate ones.

**Manuscript changes:** p21, lines 17-19: Modification; "The sparse $\rho$ map for SURF in Fig. 13, and the large spike near Fresno illustrate that while near-source surface measurements can be a powerful constraint, measurements of background concentrations provide relatively little constraints on characterizing CA anthropogenic emissions on 1-day time scales."

19. *Specific Comment: Page 22, lines 9-15: I am not sure what to make with the conclusion in the last sentence. It is true that in the cases examined in this study the aircraft flight paths and surface data might be saying different things about sources (since they are likely decoupled from one another). On other days that might not be the case when the aircraft flies in the boundary layer and will be more similar toe h surface measurements. The concluding sentence seems to contract the whole notion of using ARCTAS-CARB for their test of the adjoint technique. However, a large fraction of fires, as well as field campaign data to characterize fires are conducted over forest regions that are often located in areas of complex terrain. It seems to be a problem one has to confront.*

**AUTHOR RESPONSE:** We have revised and shortened this paragraph, removing several sentences. Cross validation is a potentially useful inversion verification technique. While it did not turn out to be very valuable in this particular case, that alone doesn't negate the value of the 4D-Var inversion, which we diagnosed using other means (posterior uncertainty estimates, DOF, etc.).

**Manuscript changes:** p22, lines 8-10: "Aircraft and surface observations do not appear to be useful for cross-validation of each other over the short timescales and limited set of flights considered here. At least for this study period, when they are not collocated, each provides some unique information to the inversion."

20. *Specific Comment: Page 22: line 27: The authors state upfront and here that the paper focused on emissions only and not other factors that can affect BC concentrations. In the future, will they extend the analysis to meteorology to reduce those factors as well?*

**AUTHOR RESPONSE:** For the results of any inversion to be meaningful, one has to consider the uncertainty introduced by imperfect meteorology. While we attempted to do this by introducing an ensemble-based model variance in **R**, that approach leaves errors in subgrid vertical mixing that can not be corrected. A more robust (and expensive) solution is to use an ensemble of data assimilations (EDA) and simultaneous DA for meteorological and chemical variables. That would be one logical next step for this work.

21. *Specific Comment: Conclusion: There are a few points I find missing or poorly artic- ulated. The first is related to recommendation on future sampling. Here and there in the paper the authors mention or allude to changes in the sapling strategy that would help in better constraining the emission. (1) It would be useful to include a summary of X ,Y, and Z they feel would be useful. (2) Second, how many more cases would be needed to have more robust estimates of the emissions? (3) Finally, another discussion I mentioned in an early comment, is the computational cost and effort of using an adjoint technique worth the cost? One could take an alternative approach is to simply perform a small number of sensitivity simulation that scale the emissions to get a better fit. This is of contrast to a brute force method, which is not as mathematically pleasing as the adjoint.*

**AUTHOR RESPONSE:** (1) We have revised the manuscript as described below.

**Manuscript changes:** p19, line 4: Removal; "Further decreasing uncertainty would require observing the same phenomena more thoroughly, either for longer periods, with greater spatial coverage, or with more instruments."

p19, lines 24-27: Addition; "That strategy is consistent with what is generally known: further decreasing uncertainty requires observing the same phenomena more thor- oughly. For hourly to daily time scales, more observations are needed close to and downwind of chemical sources, and at high spatial and temporal resolution (e.g., from repeated aircraft overpasses, extra aircraft, hourly-average surface sites, satellites)."

Also, we again revisit the topic of the types of observations that would be most useful in the Conclusions:

"...high spatial and/or temporal resolution concentration measurements from research campaigns or geostationary satellites are necessary to provide the sufficient constraints on inventory errors."

(2) 4D-Var does not provide this information – for that we would need the adjoint of the adjoint, and of the optimization routine itself. Clearly this goes beyond the scope of the present work.

(3) When comparing the computational and development costs of a Bayesian top-down method (e.g., adjoint-based 4D-Var, EnKF, or LPDM) to an ad-hoc scaling, one must consider the differences in the types of solutions that these methods provide. Indeed, a "small number of sensitivity simulations" can be used to derive low-dimension re- scalings of the emissions , as we demonstrated in Section 2.2. In contrast, a high resolution Bayesian posterior has much more spatial-temporal information than can be provided by a sensitivity study. Further, a Bayesian inversion's goal goes beyond getting "a better fit", which runs the risk of overfitting data that contains noise with- out regards for prior information or model errors; additionally, Bayesian approaches afford posterior diagnostics regarding the reduction in uncertainty and information content. This auxiliary information, not available from a simple emissions rescaling based on low-dimension sensitivity calculations, is useful for evaluating the value of the observations and the inversion result.

22. *Specific Comment: Figure 2: The colors used for the vegetation distribution make it difficult to see the red dots denoting the fires. It would be useful to include the aircraft flight paths.*

    **AUTHOR RESPONSE:** We added the 22 June flight transect, and also added white outlines to the fires to differentiate them from the surrounding colors.

    **Manuscript changes:** Replaced Figure 2.

23. *Specific Comment: Figure 7: Consider changing the color scale, it is very difficult to see changes from one figure to the next.*

    **AUTHOR RESPONSE:** We smeared out the color scale on the low end to make the large outliers stick out. Since the variation in neighboring grid cells can be quite large, it is difficult to pick out single grid cells. Also, with the difference in re-gridding between QFED and FINN, comparison between the two inventories is tenuous at grid scale. The posterior-prior increments in Figure 8 and the EA totals in Table 5 serve as other means of comparison.

24. *Specific Comment: Figure 9: The caption denotes posterior BB, but it is not clear which lines in the figure are posterior based on the legend.*

    **Manuscript changes:** Modified caption; "Hourly BB diurnal emission patterns for the four EAs and all inversion scenarios for 22 June, 00Z-23Z, with the time shown in LT. The priors are shown as black lines, while the posteriors from specific inversion scenarios are shown in color. Note that FINNv1.0 did not have any fires in EA4 on 21 June."

**2 Responses to Anonymous Referee #3**

1. *Specific Comment: One major concern of the MS is that the numerical experiments were only based on two real case forecasting. Since the atmospheric chemistry and meteorological conditions vary day to day. It is suggested that the authors extend the experiments.*

    **AUTHOR RESPONSE:** Indeed the cross validation was meant to check whether the posterior scaling factors found on one day are consistent with observations on subsequent days. They were not. We conclude that an observing strategy that targets specific source regions for longer periods of time would be more beneficial than, e.g., trying to cover the entire state of California in a single campaign. Much of this work was geared towards seeing what exactly this new tool can accomplish with a given set of observations. In future work it will be applied to many other observation sets.

2. *Specific Comment: How many numbers of the control variables in your CHEMDA system? Only BC? Please clarify the observation Operator and its adjoint. If the control variable ?same? as observation, the observation Operator and adjoint is simply as interpolation method. If not, you should clarify in the manuscript. In addition, how to deal with cross correlation between control variables (e.g. NO3 is correlate with SO4) if the number of CV is more than two.*

   **AUTHOR RESPONSE:** The observation operator includes temporal averaging and a conversion from $\mu g\, BC\, m^{-3}$ in the observation to $\mu g\, BC\, (kgdryair)^{-1}$ in the model. We describe those details and those of the model adjoint derivation in Guerrette and Henze (2015). Although BC is the only chemical species for which emissions are constrained, the sources are distributed across the entire domain. There are many zero-emission grid cells for BB sources. In total there are 5080 grid cells with non-zero emissions (# biomass burning + # anthropogenic), each with a scaling factor every hour (size(CV) = 121,920 for 24 hours) — i.e., p14, line 35.

3. *Specific Comment: Incremental method is commonly used method in data assimilation. The manuscript appear "incremental" many times. I suggest delete redundant "incremental" in the manuscript.*

   **AUTHOR RESPONSE:** While we agree that needless repetition of an established term ought to be avoided, in most cases we specifically refer to the incremental formulation to distinguish it from other approaches that previous authors have followed or to make note of particular advantages or disadvantages of this method. Here is a list of places where we feel this is the case:

   - p3, line 25
   - p4, line 14
   - p5, line 15
   - p7, line 3
   - p7, line 6
   - p8, lines 28 and 30
   - p12, line 26
   - p13, line 1
   - p22, line 12

   **Manuscript changes:** Removed "incremental" at p14, line 33. Removed "4D-Var" at p22, line 19.

4. *Specific Comment: Background error covariance (BEC) is important in data assimilation. The observation information spread to model grid cells via BEC. The authors mentioned chemical emissions heterogeneous. However, the construction of the "B" in the manuscript seems to be homogeneous?*

   **AUTHOR RESPONSE:** Yes the chemical emissions are heterogenous. We use multiplicative uncertainties through a log-normal background term to circumvent the need to use heterogenous Gaussian distributed errors. We also discussed the possibility that the relative uncertainties are spatially heterogeneous. One might derive such heterogenous uncertainties using a Monte Carlo approach, but the details of such a procedure are far beyond the scope of this work.

**3    Additional changes**

1. We modified the Damped Gauss Newton section to remove needless details. Since writing the paper, we have added an optimal line search for the damping parameter, and the new results are nearly identical to those presented in this work. Thus we also eliminated the discussion of "Heuristic Damped Gauss Newton".

2. Modification; the posterior covariance is changed from $\mathbf{A}$ to $\mathbf{P}^\mathrm{a}$ throughout for consistency with previous literature.

3. We rearranged the second paragraph (p2, lines 7-13) to improve readability.

4. "heterogenous" is corrected to "heterogeneous" throughout

5. p4, line 8: Modification; "incremental 4D-Var in WRFDA-Chem" changed to "WRFDA"

6. p5, line 34 to p6, line 2: Modification; "For temperate forests QFED scales aerosols by $\times 4.5$ throughout the world. That vegetation category accounts for 80% of the wild-fire BC in California during 22-30 June 2008. The ..."

7. p6, lines 5-6: Addition; "..., and without it FINNv1.0 and QFED would differ by $\times 10$ during the ARCTAS-CARB campaign."

8. p14, line 19; p14, line 24; p16, line 21: Addition; "relative" as in "..we use a relative grid-scale BB uncertainty.."

9. p14, line 23: Addition; "(see Sec. 3.3)"

10. p16, line 27: Modification; changed "in" to "at" as in "Meanwhile, at other times..."

11. p17, line 24: Reference Correction; "Zhang et al., 2012" fixed to "Zhang et al., 2014a"

12. p18, line 15: Removal; "and " as in "..., and additional inversions, "

13. p21, line 23: Removal; "Even before carrying out such a test, " as in "Even before carrying out such a test, the heterogeneous adjoint ..."

14. p22, line 4: Modification; "in any" changed to "derived from"

15. p22, line 27: Added reference to Song et al. (2016).